# In vitro education of human natural killer cells by KIR3DL1

Jason Pugh[1], Neda Nemat-Gorgani[1], Zakia Djaoud[1], Lisbeth A Guethlein[1], Paul J Norman[2], Peter Parham[1]

During development, NK cells are "educated" to respond aggressively to cells with low surface expression of HLA class I, a hallmark of malignant and infected cells. The mechanism of education involves interactions between inhibitory killer immunoglobulin–like receptors (KIRs) and specific HLA epitopes, but the details of this process are unknown. Because of the genetic diversity of HLA class I genes, most people have NK cells that are incompletely educated, representing an untapped source of human immunity. We demonstrate how mature peripheral KIR3DL1+ human NK cells can be educated in vitro. To accomplish this, we trained NK cells expressing the inhibitory KIR3DL1 receptor by co-culturing them with target cells that expressed its ligand, Bw4+HLA-B. After this training, KIR3DL1+ NK cells increased their inflammatory and lytic responses toward target cells lacking Bw4+HLA-B, as though they had been educated in vivo. By varying the conditions of this basic protocol, we provide mechanistic and translational insights into the process NK cell education.

## Introduction

NK cells are innate immune cells that contribute to human immunity and placentation (Parham & Moffett, 2013). Like T cells, NK cells travel throughout the body, and have the ability to lyse infected or malignant cells upon contact. Unlike T cells, however, NK cells do not require cooperation from other immune cells to mount an immune response (Greenberg & Playfair, 1974; Kiessling et al, 1976). Despite this lack of oversight, NK cells rarely harm healthy tissue or cause autoimmunity (Toubi & Vadasz, 2019). NK cells accurately target unhealthy cells in part because they sense proteins that are typically expressed on the surface of healthy cells. Among these proteins are the Class I HLA proteins, which are expressed by almost all healthy human cells (Boegel et al, 2018).

NK cells detect HLA on other cells using killer immunoglobulin–like receptors (KIRs) (Colonna & Samaridis, 1995). When an inhibitory KIR on an NK cell binds to HLA on another cell, the KIR initiates an inhibitory signal that counters activation (Valiante et al, 1996; Lanier, 2003). Malignancy and infection can each reduce a cell's expression of HLA, making that cell a target for NK cells (Seliger et al, 1997; Bukur

et al, 2012; Crux & Elahi, 2017). A missing-self response occurs when an NK cell attacks another cell because that cell does not express enough HLA.

NK cells that do not express any inhibitory receptors are hyporesponsive, likely because of the absence of the activation kinases Syk and Zap70 (Pugh et al, 2018). The magnitude of the missing-self response of KIR+ NK cells depends on whether their KIR bound HLA during cellular development. The binding of HLA by KIR during NK cell development is said to "educate" the NK cell to have a greater response if that specific HLA epitope is missing from cells it encounters thereafter (Kim et al, 2008). KIR+ NK cells that did not bind HLA during development have a comparably diminished missing-self response as a result and are called uneducated (Anfossi et al, 2006).

The KIR gene locus contains up to five inhibitory KIR genes per person (Guethlein et al, 2015). Each inhibitory KIR binds to a specific epitope on HLA (Parham & Moffett, 2013). KIR3DL2 binds the A3/11 epitope found on some HLA-A alleles. KIR3DL1 binds the Bw4 epitope of HLA-B or HLA-A. KIR2DL2 and KIR2DL3 each bind the C1 epitope, which occurs on many HLA-C and two HLA-B alleles (Moesta et al, 2008). KIR2DL1 binds the C2 epitope of HLA-C. Other inhibitory receptors expressed by NK cells include CD94:NKG2A and LILRB1, which inhibit NK cells when bound to HLA-E (Sullivan et al, 2008). Inhibitory KIR genes are switched on stochastically during NK cell development, such that individual NK cells express different numbers and combinations of KIR (Andersson et al, 2009; Schonberg et al, 2011).

Class I HLA genes are the most diverse genes in the human population, comprising thousands of HLA-A, -B, and -C alleles (Robinson et al, 2017). The diversity of KIR genes is also high (Guethlein et al, 2015; Misra et al, 2018). As a result of this combined diversity, less than 5% of the human population has a genotype in which NK cells are educated through all possible KIR-HLA interactions (Robinson et al, 2016).

Their relative safety and their ability to recognize and lyse malignant cells make NK cells an obvious choice for the development of immunotherapies. However, many NK cell immunotherapies have thus far underperformed in the clinic (Karre et al, 1986; Storkus et al, 1987; Eguizabal et al, 2014; Gras Navarro et al, 2015). One potential reason for this is that NK cells that lack education never reach their full potential. Strategies for boosting the response of NK cells to cancer include blocking KIR-HLA binding in vivo, which theoretically promotes NK cell activation (Kim & Kim, 2018). However, KIR blockade

[1]Departments of Structural Biology and Microbiology & Immunology, Stanford University School of Medicine, Stanford, CA, USA  [2]Division of Biomedical Informatics and Personalized Medicine, Department of Immunology, School of Medicine, University of Colorado Denver, Denver, CO, USA

Correspondence: peropa@stanford.edu

interventions have thus far not been successful in clinical trials. This may be partly because they are only applicable to the degree that the patient's NK cells have been educated in vivo through the targeted KIRs. By genetic chance, most individuals lack the ability to educate all of their NK cells in vivo. In this study, for the first time to our knowledge, we provide evidence that mature human peripheral NK cells expressing KIR3DL1 can be educated in vitro.

# Results

### General approach to in vitro NK cell education by KIR3DL1

To explore the possibility of achieving in vitro NK cell education, we chose to study KIR3DL1$^+$ NK cells. These cells are educated in vivo through KIR3DL1 binding to the Bw4 epitope carried by some HLA-A and B allotypes. We studied donors with uneducated KIR3DL1$^+$ NK cells, as defined by their lack of any HLA-A or HLA-B Bw4 epitope. We isolated and co-cultured donor NK cells with a target B cell line that expressed Bw4$^+$HLA-B. As a control, we also co-cultured their isolated NK cells with a target BCL that expressed only Bw4$^-$ HLA-A and -B. This first co-culture is called the training phase (Fig 1). We similarly trained NK cells from Bw4$^+$HLA-B donors. After the training phase, we added K562 target cells to each co-culture to test NK cell function. K562 are erythroleukemia cells that lack HLA class I. NK cells were then assayed for IFNγ production and degranulation. This second co-culture is called the testing phase (Fig 1). We expected that 3DL1$^+$ NK cells from Bw4$^-$ donors trained on Bw4$^+$ targets would be educated in the training phase, leading to greater IFNγ production and degranulation in the testing phase. We also anticipated that NK cells from many Bw4$^+$ donors trained on Bw4$^+$ targets would have increased IFNγ production in the testing phase

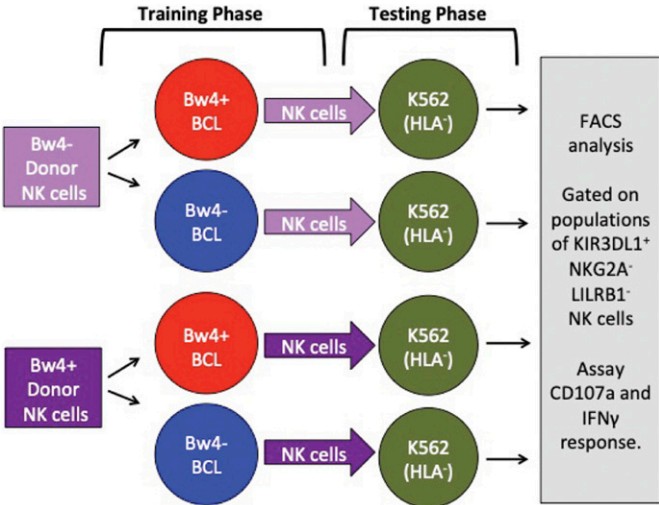

**Figure 1. General approach to in vitro NK cell education.**
In the training phase (left), NK cells isolated from either Bw4$^+$ or Bw4$^-$ donors are co-cultured with training cells that express either Bw4$^+$HLA-B (educating) or Bw4$^-$HLA-B (control). Testing lines are pretreated with etoposide VP-16. In the testing phase (right), K562 cells, which lack HLA class I, are added. The missing-self response of KIR3DL1$^+$ NK cells to the HLA$^-$ targets is then measured by IFNγ and/or degranulation assays using flow cytometry.

because the Bw4$^+$HLA-B allotypes expressed by the training cell lines have higher avidity for KIR3DL1 than most allotypes expressed by our cohort of NK cell donors.

### In vivo education by KIR3DL1 is detectable by IFNγ production, but not degranulation, after extended cell culture with recombinant human IL-2

We first tested the robustness of in vivo NK cell education using samples from our donor cohort and conditions similar to those planned for in vitro NK cell education, notably a 5-d cell culture with IL-2. Two concentrations of recombinant human IL-2 were compared: low 100 U/ml and high 500 U/ml IL-2. After the culture, NK cells were tested for a missing-self response against K562 target cells. IFNγ production and NK cell degranulation were assayed by flow cytometry. We compared the response of KIR3DL1$^+$ NK cells from Bw4$^+$ donors, which were educated in vivo by Bw4, to KIR3DL1$^+$ NK cells from Bw4$^-$ donors, which were not educated by Bw4. To reduce effects due to inhibitory receptors other than KIR, NK cells expressing NKG2A and/or LILRB1 were excluded from the analysis.

After culture in high IL-2, KIR3DL1$^+$ NK cells from Bw4$^+$ donors produced more IFNγ in response to missing-self than KIR3DL1$^+$ NK cells from Bw4$^-$ donors (Fig 2A). On average, in vivo education by Bw4 accounted for a 47% increase in the frequency of KIR3DL1$^+$ NK cells producing IFNγ (formula in the Materials and Methods section). Although in vivo education by Bw4 increased the IFNγ response, it had no effect on CD107a expression, a proxy for degranulation. This negative result was observed for NK cells cultured in high or low IL-2 (Fig 2B).

In these experiments, in vivo education mediated by KIR3DL1 was observed for the cytokine response but not the cytotoxic response. Thus, any effect of in vivo NK cell education mediated by KIR3DL1 on the cytotoxic response is no longer apparent after a 5-d cell culture with IL-2.

### KIR3DL1$^+$ NK cells from Bw4$^-$ donors exhibit increased missing-self response after in vitro education with a Bw4$^+$ BCL

To educate KIR3DL1$^+$ NK cells in vitro, NK cells isolated from Bw4$^+$ to Bw4$^-$ donors were co-cultured with one of two EB- transformed B cell lines (BCL), either a Bw4$^+$BCL or a Bw4$^-$BCL. Both BCLs express similar amounts of cell surface HLA class I (Fig S1A). Training co-cultures were performed in either low or high IL-2 conditions. After 5 d of co-culture, NK cells were tested for a missing-self response against K562 target cells in high IL-2 medium. IFNγ production and degranulation were assayed by flow cytometry. NK cells expressing NKG2A and/or LILRB1 were excluded from analysis.

For NK cells cultured in low IL-2, KIR3DL1$^+$ NK cells from Bw4$^-$ donors gained a significant increase in their IFNγ response to missing-self when trained with Bw4$^+$BCL, compared with training with Bw4$^-$BCL (Fig 3A). On combining data from three experiments representing 25 Bw4$^-$ donors, in vitro education by Bw4$^+$BCL in low IL-2 accounted for a 38% increase in the frequency of IFNγ$^+$ cells (Table 1). In a combined analysis of 20 Bw4$^+$ donors, in vitro education by Bw4$^+$BCL in low IL-2 accounted for a 68% increase in the frequency of IFNγ$^+$ cells (Table 1). However, the effect of in vitro education in low IL-2 on NK cells from Bw4$^+$ donors did not reach

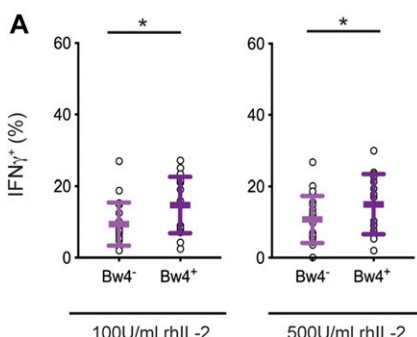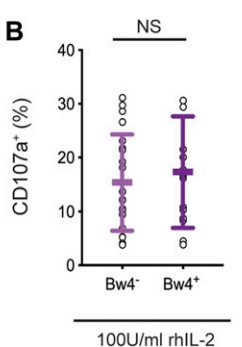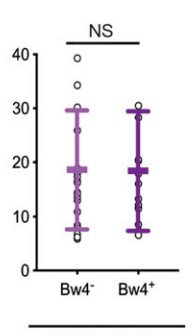

**Figure 2. After 5 d of in vitro cell culture, NK cell education is detectable by IFNγ production but not degranulation.**
NK cells isolated from PBMCs were cultured in medium with rhIL-2 at the concentrations indicated. After 5 d, the K562 cells were added at a 10:1 E:T ratio, in medium with 500 U/ml rhIL-2 and anti-CD107a. 6 h later, the NK cells were stained and analyzed by flow cytometry. Shown are the mean ± SD of three experiments combined totaling 21 Bw4⁻ donors and 18 Bw4⁺ donors. **(A)** Frequency of IFNγ⁺ cells in the viable KIR3DL1⁺NKG2A⁻LILRB1⁻ NK cell gate. Shown are the results of a *t* test. *$P < 0.05$. **(B)** Frequency of CD107a⁺ cells in the viable KIR3DL1⁺NKG2A⁻LILRB1⁻ NK cell gate. Shown are the results of a *t* test. NS = not significant. Source data are available for this figure.

statistical significance unless data from multiple experiments were combined (compare Fig 3A with Table 1).

Among NK cells cultured in high IL-2, KIR3DL1⁺ NK cells from Bw4⁻ donors gained a significant increase in their IFNγ response to missing-self when trained with Bw4⁺BCL, compared with training with Bw4⁻BCL (Fig 3A). Training with Bw4⁺BCL in high IL-2 resulted in a 33% increase in the frequency of IFNγ⁺ cells (Table 1). In vitro education by Bw4⁺BCL in high IL-2 accounted for an 85% increase in the frequency of IFNγ⁺ cells among 20 Bw4⁺ donors (Table 1). However, the effect of Bw4⁺BCL training on NK cells from Bw4⁺ donors was statistically less significant than comparable training of NK cells from Bw4⁻ donors (Fig 3A and Table 1).

Training with Bw4⁺BCL in high or low IL-2 did not affect the frequency of CD107a⁺ cells for either Bw4⁺ donors or Bw4⁻ donors (Fig 3B and Table 1). Because no in vivo NK cell education was detectable by degranulation after similar culture conditions (Fig 2B), this result is consistent with KIR3DL1⁺ cells having achieved in vitro NK cell education.

### KIR3DL1⁺ NK cells trained with Bw4⁺BCL and rhIL-12 exhibit enhanced degranulation in response to missing-self

To assess the effect of inflammatory cytokines on in vitro education, we trained NK cells with either Bw4⁺BCL or Bw4⁻BCL in medium containing both high IL-2 and 50 ng/ml recombinant human IL-12, abbreviated as IL-12. Culturing NK cells in IL-12 dramatically increased

the frequency of IFNγ⁺ NK cells that responded to missing-self, resulting in >60% of the KIR3DL1⁺ NK cells producing IFNγ (compare Fig 3A with Fig 4A). However, training with Bw4⁺BCL and IL-12 did not increase the frequency of IFNγ⁺ cells over that obtained from training with Bw4⁻BCL and IL-12. A similar negative result was observed for NK cells from Bw4⁻ to Bw4⁺ donors (Fig 4A).

We next compared the degranulation of IL-12–trained KIR3DL1⁺ NK cells in response to K562 cells. Among Bw4⁻ donors, the frequency of CD107a⁺ cells increased by 22% when trained with Bw4⁺BCL and IL-12, compared with training with Bw4⁻BCL and IL-12 (Fig 4B). Training KIR3DL1⁺ NK cells from Bw4⁺ donors with Bw4⁺BCL and IL-12 did not improve their degranulation (Fig 4B).

Next, we measured the frequency of NK cells that produced both IFNγ and degranulated. KIR3DL1⁺ NK cells that were trained with Bw4⁺BCL and IL-12 were 21.0% ± 8.0% CD107a⁺IFNγ⁺ on average (Fig 4C). This was significantly greater than that achieved by KIR3DL1⁺ NK cells trained with Bw4⁻BCL, of which only 16.1% ± 8.0% were CD107a⁺IFNγ⁺ (Fig 4C). A statistically more significant difference was observed between KIR3DL1⁺ NK cells trained with Bw4⁻BCL and Bw4⁺BCL, when both degranulation and IFNγ production were considered (compare Fig 4B to Fig 4C).

In summary, exposing NK cells to IL-12 overrides the effect of in vitro education on IFNγ production. However, IL-12 preserves the effect of in vitro education on degranulation, which was unaffected by training with IL-2 alone. Moreover, because almost all IFNγ⁺ NK cells also degranulated in response to missing-self when trained

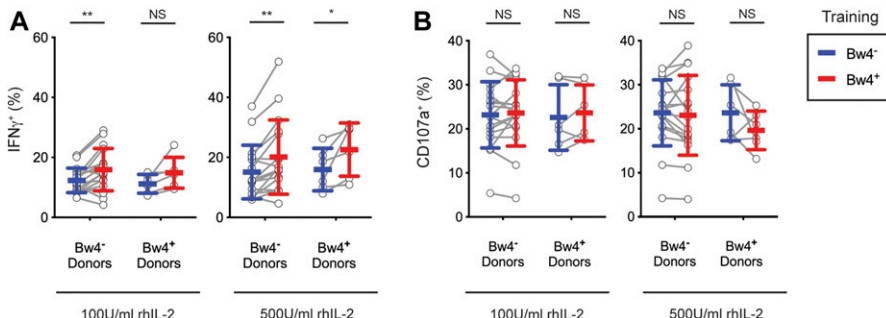

**Figure 3. In vitro training of KIR3DL1⁺ cells with Bw4⁺BCL improves the missing-self response by IFNγ production but not degranulation.**
NK cells were isolated from the PBMCs of 17 Bw4⁻ donors and six Bw4⁺ donors. NK cells were then co-cultured with either Bw4⁺BCL or Bw4⁻BCL at an 8:3 E:T ratio in medium with rhIL-2 at the concentrations indicated. After 5 d, K562 cells were added at a 10:1 E:T ratio, in medium with 500 U/ml rhIL-2 and anti-CD107a. 6 h later, NK cells were stained and analyzed by flow cytometry. All panels are representative of at least three replicate experiments. Mean ± SD are shown. **(A)** Frequency of IFNγ⁺ cells in the viable KIR3DL1⁺NKG2A⁻LILRB1⁻ NK cell gate. Shown are the results of a Sidak's multiple comparison test from

paired two-way ANOVAs. **$P < 0.01$, *$P < 0.05$. **(B)** Frequency of CD107a⁺ cells in the viable KIR3DL1⁺NKG2A⁻LILRB1⁻ NK cell gate. Shown are the results of a Sidak's multiple comparison test from a paired two-way ANOVA. NS = not significant.
Source data are available for this figure.

**Table 1.   Results of in vitro training with Bw4 on the missing–self response of KIR3DL1⁺ NK cells.**

| Donor Allotype | | Bw4⁻ | | | | % Change (IFNγ) | % Change (CD107a) | Bw4⁺ | | | | % Change (IFNγ) | % Change (CD107a) |
|---|---|---|---|---|---|---|---|---|---|---|---|---|---|
| Training Cell Allotype | | Bw4⁻ | | Bw4⁺ | | | | Bw4⁻ | | Bw4⁺ | | | |
| Type of Response to K562 | | IFNγ⁺ (%) | CD107a⁺ (%) | IFNγ⁺ (%) | CD107a⁺ (%) | | | IFNγ⁺ (%) | CD107a⁺ (%) | IFNγ⁺ (%) | CD107a⁺ (%) | | |
| Training conditions | Low IL2 | 10.5 | 22.4 | 14.5 | 22.9 | **38.1\*\*** | 2.2 | 6.9 | 20.7 | 11.6 | 19.1 | **68.1\*\*** | −7.7 |
| | High IL2 | 14.8 | 19.8 | 19.7 | 18.7 | **33.1\*\*** | −5.6 | 11.9 | 15.9 | 22.0 | 14.8 | **84.8\*** | −6.9 |

NK cells were co-cultured with either Bw4⁺BCL or Bw4⁻BCL in medium with either 100 U/ml rhIL-2 or 500 U/ml rhIL-2. After 5 d, K562 cells were added at a 10:1 E:T ratio, in medium with 500 U/ml rhIL-2 and anti-CD107a. 6 h later, the NK cells were stained and analyzed by flow cytometry. Combined data from three experiments, totaling 25 Bw4⁻ donors and 20 Bw4⁺ donors, are shown. The frequency of either CD107a⁺ or IFNγ⁺ NK cells in the viable KIR3DL1⁺NKG2A⁻LILRB1⁻ NK cell gate is shown. Percent change was calculated as: ([Bw4⁺ training – Bw4⁻ training]/Bw4⁻ training). Bold type indicates a significant statistical result as assessed by Sidak's multiple comparison test from paired two-way ANOVAs. Asterisks denote the degree of significance: \*\*$P < 0.01$, \*$P < 0.05$.

with IL-12, in vitro education with IL-12 maximizes the overall missing-self response of KIR3DL1⁺ NK cells.

### Bw4⁺BCL training does not cause differences in the division, survival, or activation of NK cells that can explain their increased response to missing-self

We next considered scenarios other than education that could explain the increased response to missing-self observed for NK cells after Bw4⁺BCL training. Because NK cells were cultured with training cells for days before assessment of their missing-self response, one such scenario would be if more KIR3DL1⁺ NK cells survived co-culture

with Bw4⁺BCL cells than co-culture with Bw4⁻BCL cells. To assess this possibility, the viability and KIR3DL1 expression of NK cells after 5 d of training were assessed. The frequency of viable KIR3DL1⁺ NK cells after co-culture with Bw4⁺BCL cells was not significantly greater than those of KIR3DL1⁺ NK cells co-cultured with Bw4⁻BCL cells (Fig S1C). The frequency of NK cells that divided during co-culture, as measured by bromodeoxyuridine incorporation, was no higher for NK cells trained with Bw4⁺BCL than for those trained with Bw4⁻BCL (Fig S1D).

Another scenario that could yield false evidence for in vitro education would be if Bw4⁺BCL expressed more ligands for NK cell activation than Bw4⁻BCL. If this was the case, NK cells lacking KIR3DL1 should also show an increased missing-self response after

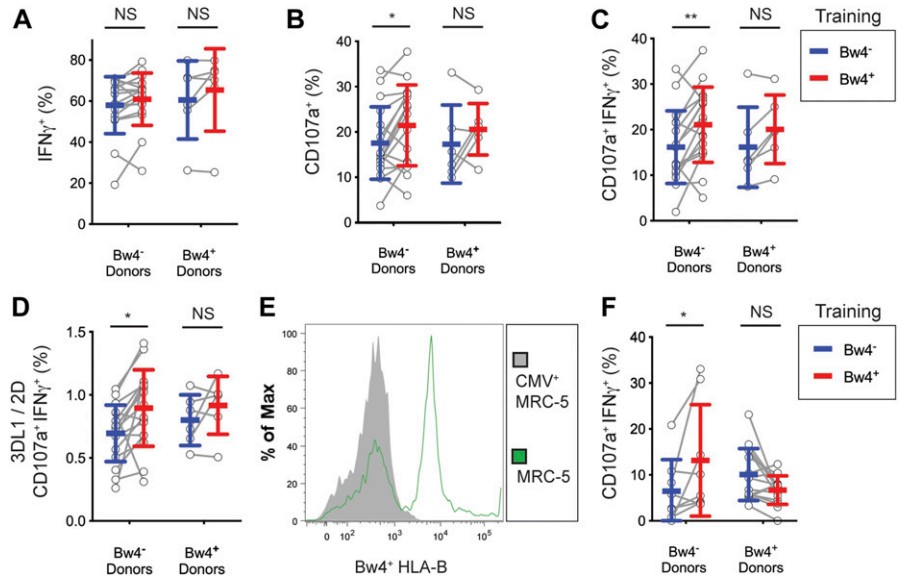

**Figure 4.   In vitro training with Bw4⁺BCL and rhIL-12 maximizes the missing-self response of KIR3DL1⁺ NK cells.**
For **(A, B, C, D)**, NK cells were isolated from the PBMCs of 17 Bw4⁻ donors and six Bw4⁺ donors. NK cells were then co-cultured with either Bw4⁺BCL or Bw4⁻BCL at an 8:3 E:T ratio in medium with 500 U/ml rhIL-2 and 50 ng/ml rhIL-12. After 5 d, K562 cells were added at a 10:1 E:T ratio, in medium with 500 U/ml rhIL-2 and anti-CD107a. 6 h later, NK cells were stained and analyzed by flow cytometry. **(A, B, C, D)** are representative of at least three replicate experiments. Mean ± SD are shown. **(A)** Frequency of IFNγ⁺ cells in the viable KIR3DL1⁺NKG2A⁻LILRB1⁻ NK cell gate. Shown are the results of a Sidak's multiple comparison test from a paired two-way ANOVA. **(B)** Frequency of CD107a⁺ cells in the viable KIR3DL1⁺NKG2A⁻LILRB1⁻ NK cell gate. Shown are the results of a Sidak's multiple comparison test from a paired two-way ANOVA. \*$P < 0.05$. **(C)** Frequency of IFNγ⁺CD107a⁺ cells in the viable KIR3DL1⁺NKG2A⁻LILRB1⁻ NK cell gate. Shown are the results of a Sidak's multiple comparison test from a paired two-way ANOVA. \*\*$P < 0.01$. **(D)** Frequency of IFNγ⁺CD107a⁺ cells in the viable KIR3DL1⁺NKG2A⁻LILRB1⁻ NK cell gate divided by the frequency of IFNγ⁺CD107a⁺ cells in the viable KIR3DL1⁻KIR2D⁺NKG2A⁻LILRB1⁻ NK cell gate. Shown are the results of a Sidak's multiple comparison test from a paired two-way ANOVA. \*$P < 0.05$. **(E)** MRC-5 cells were either co-cultured alone or with human CMV for 4 d. MRC-5 cells were then isolated and stained with anti-Bw4+HLA-B antibody and analyzed by flow cytometry. Viable MRC-5 cells are shown. **(F)** NK cells isolated from PBMCs were co-cultured with either Bw4⁺BCL or Bw4⁻BCL at an 8:3 E:T ratio. After 5 d, CMV-infected MRC-5 cells were added at a 10:1 E:T ratio. 6 h later, the NK cells were stained and analyzed by flow cytometry. Shown are the combined results of two experiments, representing three replicate experiments. The frequency of IFNγ⁺CD107a⁺ cells in the viable KIR3DL1⁺NKG2A⁻LILRB1⁻ NK cell gate are shown. Shown are the results of a Sidak's multiple comparison test from a paired two-way ANOVA. NS = not significant. \*$P < 0.05$.
Source data are available for this figure.

training with Bw4+BCL. To test this possibility, we normalized the missing-self response of KIR3DL1+ NK cells to that of pan-KIR2D+KIR3DL1− NK cells. These cells express at least one KIR and have, therefore, reached a similar stage of maturity as KIR3DL1+ cells, but are unaffected by interactions with Bw4+HLA-B. Among NK cells trained with Bw4+BCL, the 3DL1/pan-2D ratio of CD107a+IFNγ+ NK cells responding to missing-self was 0.90 ± 0.30, a significantly higher ratio than the 0.70% ± 0.22% gained by Bw4−BCL training (Fig 4D). This result confirms that the improvement conferred by Bw4+BCL training is specific to KIR3DL1+ NK cells. Therefore, the increased response to missing-self that we observed in NK cells after Bw4+BCL training was not due to an abundance of NK cell activation ligands on the Bw4+BCL compared with the Bw4−BCL.

### After training with Bw4+BCL, KIR3DL1+ NK cells from Bw4− individuals respond better to missing-self induced by CMV infection

We next tested if training KIR3DL1+ NK cells with a Bw4+BCL could enhance their response to target cells that had lost HLA expression due to a viral infection. We chose the human fibroblast line MRC-5 as a viral host because one of its HLA-B allotypes has the Bw4 epitope (Tabi et al, 2001). We infected MRC-5 cells with human CMV because CMV uses immunoevasins (Babic et al, 2010) to downregulate HLA-A and HLA-B from the surface of MRC-5 cells (Ameres et al, 2013). We, therefore, predicted that KIR3DL1+ NK cells educated with Bw4 should respond more aggressively to CMV-infected MRC-5 cells than KIR3DL1+ NK cells lacking education with Bw4.

After infection with CMV, MRC-5 cells down-regulated their surface expression of HLA-B, as measured by an antibody targeting the Bw4 epitope of HLA-B (Fig 4E). Among KIR3DL1+ NK cells from Bw4− donors, 13.2% ± 12.1% of those trained with the Bw4+BCL responded to CMV-infected MRC-5 cells by degranulation and production of IFNγ (Fig 4F). This was significantly more cells than the 6.5% ± 6.9% degranulation and IFNγ production achieved by KIR3DL1+ NK cells trained with the Bw4−BCL. KIR3DL1+ NK cells from Bw4+ donors showed no improvement in their responses to CMV-infected MRC-5 cells because of training with Bw4+BCL (Fig 4F). We conclude that in vitro education through KIR3DL1+ improves the response to missing-self, induced by CMV infection.

### In vitro education of KIR3DL1+ NK cells by Bw4+HLA-B depends on the interaction of KIR3DL1 with Bw4+HLA-B

The NK cells of some donors showed no improvement after training with Bw4+BCL (Fig 4C). NK cell education depends upon the strength of the interaction between KIR3DL1 and HLA-B, which varies with both the KIR3DL1 and HLA-B allotypes (Carr et al, 2005; Yawata et al, 2006; Boudreau et al, 2016). Thus, donor variation in KIR3DL1 and/or allotype could explain these differences in the efficacy of training. To examine this possibility, we calculated a relative binding and expression factor (B&EF) using the KIR3DL1 and HLA-B types of the donors as well as the HLA-B types of the Bw4+BCL (see the Materials and Methods section). The B&EF value reflects the strength of the KIR3DL1-Bw4 interaction during training, as well as the average surface expression of KIR3DL1 for each donor. The increased response to missing-self due to Bw4+BCL training was then calculated

as the difference in the frequency of CD107a+IFNγ+ NK cells after training with Bw4+BCL or Bw4−BCL (data shown in Fig 4C).

A higher B&EF was positively correlated with greater increase in the missing-self response due to in vitro education (Fig 5A). This correlation was significant (P < 0.004) and had an $r^2$ of 0.54. This result suggests that the number and strength of the bonds between KIR3DL1 on NK cells and HLA-B on the Bw4+BCL determines the efficacy of in vitro education.

We further tested whether the improved missing-self response caused by Bw4+BCL training depended on the binding of KIR3DL1 to the Bw4+HLA-B present in the training phase. To do this, we compared in vitro education by Bw4+BCL in the presence and absence of F(ab')$_2$ fragments derived from W6/32, an HLA class I–specific monoclonal antibody (Fig S2A). In the presence of W6/32 F(ab')$_2$ there was no significant missing-self response that resulted from culture with either Bw4+BCL or Bw4−BCL (Fig 5B). In contrast, in the absence of W6/32 F(ab')$_2$, Bw4+BCL training led to a significantly higher missing-self response than Bw4−BCL training (Fig 5B).

These results show that the binding of KIR3DL1 expressed on NK cells to Bw4+HLA-B expressed on BCL cells during the training phase is responsible for the enhanced missing-self response we observed among KIR3DL1+ NK cells.

### In vitro education of KIR3DL1+ NK cells by a B cell line expressing Bw4+HLA-B

We tested if in vitro NK cell education could be achieved by a single Bw4+HLA-B allotype. To do this, the 721.221 cell line, which has no endogenous expression of HLA class I, was transfected to express HLA-B*58:01 that has the Bw4 epitope (221Bw4+). As well as this 221Bw4+ transfectant, a control transfectant (221Bw4−) was made that expresses HLA-B*08:01 and lacks the Bw4 epitope. The HLA-B*08:01 and HLA-B*58:01 constructs used for transfection were mutated at critical positions to ensure that the nonamer peptides derived from the leader sequence could not be bound by HLA-E (Michaelsson et al, 2002).

After training by the 221Bw4+ transfectant, KIR3DL1+ NK cells from Bw4− donors exhibited a greater missing-self response than that achieved by training KIR3DL1+ NK cells with the 221Bw4− transfectant (Fig 5C). That such education was achieved alone by Bw4+HLA-B suggests that in vitro education of KIR3DL1+ NK cells is not dependent on any inhibitory receptor other than KIR3DL1.

In preliminary experiments, we found that if 721.221 training cells were not treated with VP-16 etoposide before education, viable Bw4+ training cells remained present during the testing phase, where they inhibited the response of KIR3DL1+ NK cells to K562 target cells (Fig S2C). Subsequently, we ensured that all the training cells had been killed before beginning the effector phase.

### In vitro education of KIR3DL1+ NK cells requires training by the intact Bw4 epitope

We next investigated whether an intact Bw4 epitope was required to educate KIR3DL1+ NK cells in vitro. Previous analysis (Sanjanwala et al, 2008) showed that replacement of leucine 82 with arginine (mutant L82R) abrogated the Bw4 epitope of HLA-B*51:01, whereas replacement of isoleucine 80 with asparagine (mutant I80N) mutant

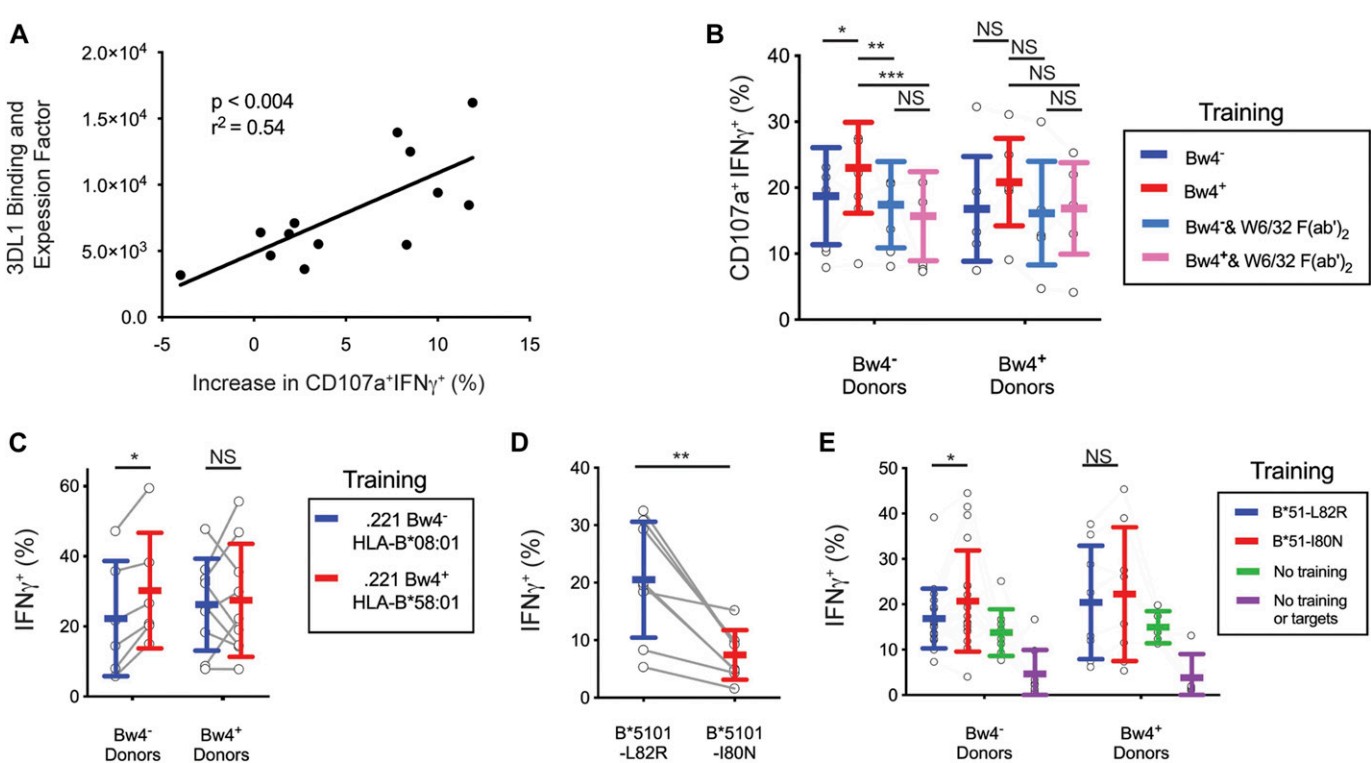

**Figure 5. The increased missing-self response of NK cells trained with Bw4$^+$BCL is dependent on the binding of KIR3DL1 to HLA.**
**(A)** Y-axis: AB&EF was assigned to each of 13 donors based on published KIR3DL1-HLA-B binding data (Boudreau et al, 2016), the HLA-B genotypes of Bw4$^+$BCL, and the surface KIR3DL1 expression for each donor's NK cells as measured by mean fluorescence intensity (MFI). X-axis: Difference between Bw4$^+$BCL and Bw4$^-$BCL training in the frequency of IFNγ$^+$CD107a$^+$ cells responding to missing-self, as presented in Fig 4D. Shown is the linear regression with the significance of non-zero slope and the goodness of fit calculation. These data are representative of at least three replicate experiments. **(B)** Bw4$^+$BCL and Bw4$^-$BCL were treated with F(ab')$_2$ fragments of the anti-HLA antibody W6/32. NK cells were isolated from PBMCs and co-cultured with either treated or untreated Bw4$^+$BCL or Bw4$^-$BCL. Cells were co-cultured at an 8:3 E:T ratio in medium with 500 U/ml rhIL-2 and 50 ng/ml rhIL-12. After 5 d, K562 cells were added at a 10:1 E:T ratio. 6 h later, the NK cells were stained and analyzed by flow cytometry. Shown is the frequency of IFNγ$^+$CD107a$^+$ cells in the viable KIR3DL1$^+$NKG2A$^-$LILRB1$^-$ NK cell gate. Shown are the combined results of two experiments comprising 11 Bw4$^-$ donors and 7 Bw4$^+$ donors. These results represent at least three replicate experiments. Shown are mean ± SD and the results of a Sidak's multiple comparison test from a paired two-way ANOVA. *$P$ < 0.05, **$P$ < 0.01, ***$P$ < 0.001. **(C)** NK cells were isolated from PBMCs and co-cultured with 721.221 cells transfected to express either HLA-B*58:01 (Bw4$^+$) or B*08:01 (Bw4$^-$). The cells were co-cultured at an 8:3 E:T ratio in medium with 500 U/ml rhIL-2. After 3 d, the K562 cells were added at a 10:1 E:T ratio. 6 h later, the NK cells were stained and analyzed by flow cytometry. The frequency of IFNγ$^+$ cells in the KIR3DL1$^+$ gate is shown. Shown are combined data from two experiments comprising six Bw4$^-$ donors and nine Bw4$^+$ donors. These data represent at least three replicate experiments. Shown are mean ± SD and the results of a Sidak's multiple comparison tests from a paired two-way ANOVA. *$P$ < 0.05. **(D)** 721.221 cells were transfected to express HLA-B*51:01 mutated to either abolish the Bw4 epitope (L82R, Bw4$^-$), or leave it intact (I80N, Bw4$^+$). The NK cells were isolated from PBMCs and co-cultured with either Bw4$^+$I80N or Bw4$^-$L82R in medium with 500 U/ml rhIL-2. 6 h later, the NK cells were stained with antibodies and analyzed by flow cytometry. The frequency of IFNγ$^+$ cells in the KIR3DL1$^+$ gate is shown. Shown are mean ± SD and the results of a $t$ test. **$P$ < 0.01. **(E)** NK cells were isolated from PBMC and co-cultured with 721.221 cells transfected to express HLA-B*51:01 with either the I80N (Bw4$^+$) or L82R (Bw4$^-$) mutation. Cells were co-cultured at an 8:3 E:T ratio in medium with 500 U/ml rhIL-2. Additional isolated NK cells were cultured alone in medium with 500 U/ml rhIL-2 (No training). After 3 d, K562 cells were added to co-cultures at a 10:1 E:T ratio, except for a portion of the NK cells cultured alone, which received only medium (No training or targets). 6 h later, NK cells were stained and analyzed by flow cytometry. The frequency of IFNγ$^+$ cells in the KIR3DL1$^+$NKG2A$^-$ gate is shown. Shown are the combined data comprising 20 Bw4$^-$ donors and 8 Bw4$^+$ donors from two experiments. These data represent at least three replicate experiments. Shown are mean ± SD and the results of a Sidak's multiple comparison tests from a paired two-way ANOVA. NS = not significant. *$P$ < 0.05.
Source data are available for this figure.

had no effect on the Bw4 epitope. Although functionally distinct, the L82R and I80N mutations mutants gave similar levels of cell surface expression (Fig S2B).

L82R and I80N mutants of *HLA-B*15:01* were individually transfected into 721.221 cells. A 6-h co-culture of KIR3DL1$^+$ NK cells with Bw4$^+$I80N expressing target cells produced significantly less IFNγ than KIR3DL1$^+$ NK cells co-cultured with Bw4$^-$L82R (Fig 5D). These results show that the Bw4$^+$I80N mutant of HLA-B*15:01 engages KIRDL1 to inhibit NK cell cytolysis, whereas Bw4$^-$L82R does not.

After training with Bw4$^+$I80N, KIR3DL1$^+$ NK cells from Bw4$^-$ donors had a greater missing-self response than that achieved by KIR3DL1$^+$

NK cells trained with Bw4$^-$L82R (Fig 5E). These results show that an intact Bw4 epitope is necessary for in vitro education of KIR3DL1$^+$ NK cells.

### In vitro NK cell education is not abrogated by exposure to targets lacking HLA class I

Educated mouse NK cells can lose their education if they are transplanted into mice lacking MHC class I (Wu & Raulet, 1997; Joncker et al, 2010). Given this precedent, we tested whether the missing-self response of in vitro–educated KIR3DL1$^+$ NK cells is affected by exposure to human cells lacking HLA class I.

We co-cultured one set of NK cells with Bw4⁻BCL and two sets of NK cells with Bw4⁺BCL in the presence of high IL-2. After 5 d of culture, NK cells were tested for a missing-self response to K562 cells, which express no HLA class I. NK cells co-cultured with Bw4⁻BCL and one of the sets cultured with Bw4⁺BCL were then assayed by flow cytometry. The other set of NK cells co-cultured with Bw4⁺BCL remained in culture for another day, after which they were tested for a second time with K562 cells.

Among KIR3DL1⁺ NK cells from Bw4⁻ donors trained with Bw4⁺BCL, the missing-self response during the second testing phase was not significantly lower than that observed during the first testing phase (Fig 6A). These data show that the missing-self response does not decline after a single exposure to HLA⁻ target cells.

We next tested whether in vitro education is retained after multiple rounds of exposure to target cells lacking HLA class I. Isolated NK cells from Bw4⁻ donors were co-cultured with Bw4⁻BCL and Bw4⁺BCL in the presence of high IL-2. After 5 d of training, NK cells were subjected to three rounds of testing with K562 cells, with 1 d of rest between each round. Of the KIR3DL1⁺ NK cells trained with Bw4⁺BCL in the third round of testing, 27.2% ± 3.8% of them produced IFNγ in response to K562 cells (Fig 6B). This response was significantly higher than the 24.3% ± 3.3% achieved by KIR3DL1⁺ NK cells trained with Bw4⁻BCL (Fig 6B). These data show that in vitro education of human NK cells can be retained after multiple rounds of exposure to target cells lacking HLA class I.

In general, education by Bw4⁺BCL produced a 33% increase in the frequency of IFNγ⁺ KIR3DL1⁺ NK cells, as assessed in the first testing phase (Table 1). Thus, the 13% increase observed after the third testing phase (Fig 6B) indicates that the effects of in vitro education

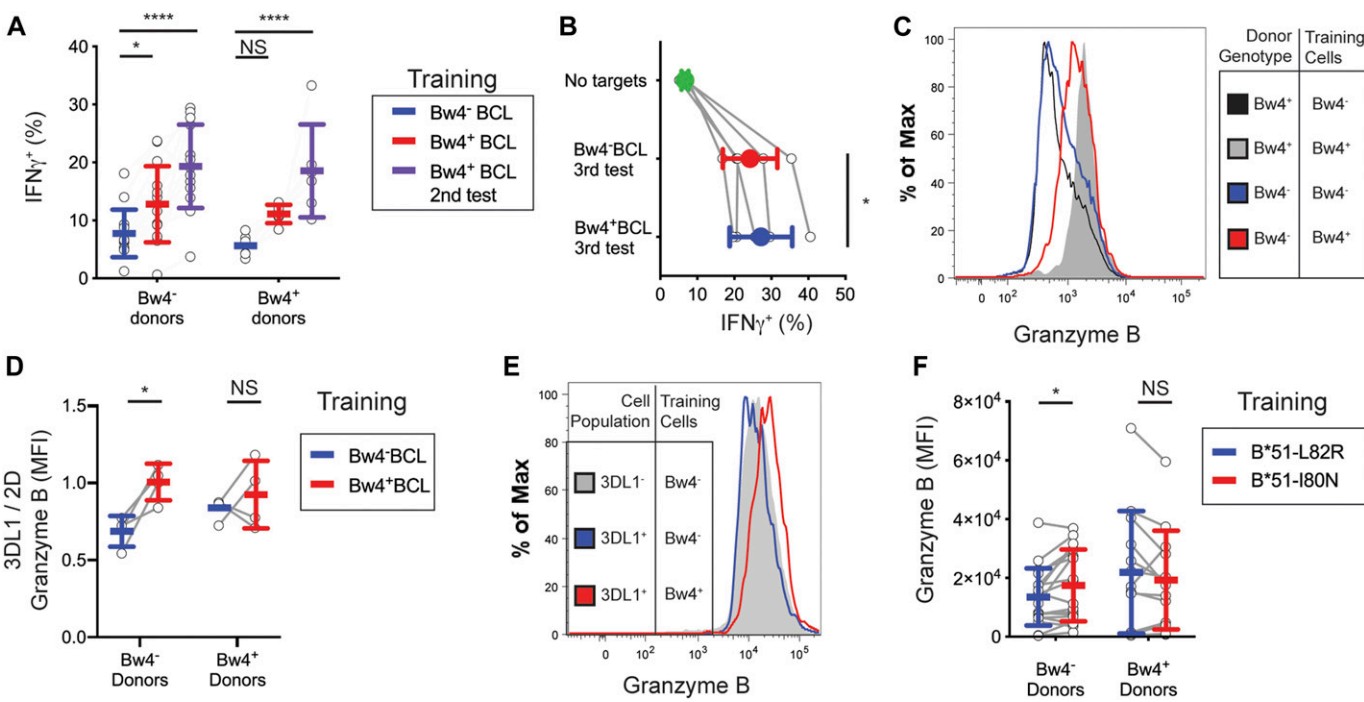

**Figure 6. Training NK cells from Bw4⁻ donors with Bw4⁺BCL improves their missing-self response to multiple rounds of HLA⁻ cells, and increases their expression of granzyme B.**

**(A)** NK cells were isolated from the PBMCs of 13 Bw4⁻ donors and six Bw4⁺ donors. Two replicate groups of NK cells were then co-cultured with Bw4⁺BCL and one group with Bw4⁻BCL, at an 8:3 E:T ratio in medium with 500 U/ml rhIL-2. After 5 d, K562 cells were added at a 10:1 E:T ratio (first testing). 6 h later, NK cells from co-cultures with Bw4⁻BCL and one of the Bw4⁺BCL co-cultures were stained and analyzed by flow cytometry. The other Bw4⁺BCL co-culture was rested for 24 h, after which more K562 cells were added at a 10:1 E:T ratio (second testing). 6 h later, NK cells were stained and analyzed by flow cytometry. The frequency of IFNγ⁺ cells in the viable KIR3DL1⁺NKG2A⁻LILRB1⁻ NK cell gate is shown. These data represent at least three replicate experiments. Shown are mean ± SD and the results of a Sidak's multiple comparison tests from a paired two-way ANOVA. *P < 0.05, ****P < 0.0001. **(B)** NK cells were isolated from the PBMCs of five Bw4⁻ donors and co-cultured with either Bw4⁺BCL or Bw4⁻BCL at an 8:3 E:T ratio in medium with 500 U/ml rhIL-2 (day 1). On days 5, 7, and 9, K562 cells were added at a 10:1 E:T ratio in medium with 500 U/ml rhIL-2. 6 h after adding K562 cells on day 9, NK cells were stained and analyzed by flow cytometry. The frequency of IFNγ⁺ cells in the viable KIR3DL1⁺NKG2A⁻LILRB1⁻ NK cell gate is shown. Shown are mean ± SD and the results of a paired t test. *P < 0.05. **(C, D)** NK cells were isolated from the PBMCs of four Bw4⁻ donors and four Bw4⁺ donors. NK cells were then co-cultured with either Bw4⁺BCL or Bw4⁻BCL at an 8:3 E:T ratio in medium with 500 U/ml rhIL-2. After 5 d, K562 cells were added at a 10:1 E:T ratio, in medium with 500 U/ml rhIL-2. Two days later, the NK cells were stained and analyzed by flow cytometry. **(C)** Shown is a histogram of granzyme B expression of cells in the viable KIR3DL1⁺NKG2A⁻LILRB1⁻ NK cell gate, obtained from concatenated flow cytometry data files for all eight donors. **(D)** Shown is the ratio between the granzyme B MFI for cells in the KIR3DL1⁺NKG2A⁻LILRB1⁻ NK cell gate and the cells in the KIR2D⁺KIR3DL1⁻NKG2A⁻LILRB1⁻ NK cell gate. Shown are mean ± SD and the results of a Sidak's multiple comparison tests from a paired two-way ANOVA. *P < 0.05. **(E, F)** NK cells were isolated from PBMCs and co-cultured with 721.221 cells transfected to express HLA-B*51:01 with either the I80N (Bw4⁺) or L82R (Bw4⁻) mutation. The cells were co-cultured at an 8:3 E:T ratio in medium with 500 U/ml rhIL-2. After 5 d, K562 cells were added at a 10:1 E:T ratio. After 2 more days, NK cells were stained and analyzed by flow cytometry. **(E)** Shown is a histogram of granzyme B expression for viable cells in either the KIR3DL1⁺NKG2A⁻LILRB1⁻ NK cell gate or the KIR3DL1⁻NKG2A⁻LILRB1⁻ NK cells gate. These data were obtained from concatenated flow cytometry data files from seven Bw4⁻ donors to five Bw4⁺ donors. **(F)** Shown is the granzyme B expression as measured by MFI of cells in the viable KIR3DL1⁺NKG2A⁻LILRB1⁻ NK cell gate. These data were obtained from 18 Bw4⁻ donors to 13 Bw4⁺ donors from three replicate experiments. Shown are mean ± SD and the results of a Sidak's multiple comparison tests from a paired two-way ANOVA. NS = not significant. *P < 0.05.

Source data are available for this figure.

decline if KIR3DL1$^+$ NK cells have repeated encounters with HLA class I negative target cells.

### NK cells educated in vitro increase their intracellular granzyme B

A distinguishing phenotype of educated NK cells is an increased amount of lysosomal granzyme B compared with uneducated NK cells (Goodridge et al, 2019). In contrast, we found no difference in lysosomal granzyme B, when KIR3DL1$^+$ NK cells from Bw4$^-$ donors were trained with Bw4$^+$BCL or Bw4$^-$BCL (Fig S2D).

This result suggests that acquisition of granzyme B by in vitro educated KIR3DL1$^+$ NK cells requires further differentiation. We examined if KIR3DL1$^+$ NK cells increase their granzyme B after training, testing, and a period of rest in which they are incubated undisturbed. The NK cells were trained with either Bw4$^+$BCL or Bw4$^-$BCL for 5 d, then tested with K562 cells and rested for 2 d in the presence of high IL-2. The KIR3DL1$^+$ NK cells from Bw4$^-$ had increased granzyme B (MFI of 1734) compared with KIR3DL1$^+$ NK cells educated by Bw4$^-$BCL (MFI of 1099) (Fig 6C).

Our study included only donors having similar levels of NK cell education mediated by the interactions of HLA-C with KIR2D (see the Materials and Methods section). We, therefore, hypothesized that in vitro education with Bw4 should increase the granzyme B in KIR3DL1$^+$ NK cells relative to KIR2D$^+$NK cells. The KIR3DL1/KIR2D ratio of granzyme B, as assessed by MFI, was 1.0 ± 0.1 MFI for NK cells from Bw4$^-$ donors educated by Bw4$^+$BCL (Fig 6D). This was a significant increase over the 0.69 ± 0.1 ratio of NK cells trained with Bw4$^-$BCL (Fig 6D).

We next determined whether NK cells educated by a Bw4$^+$221 cell line would also develop more granzyme B. NK cells were co-cultured with either Bw4$^+$I80N or Bw4$^-$L82R cells. After 3 d, K562 cells were added at a 10:1 E:T ratio, and then NK cells were rested for 2 more days. Among the KIR3DL1$^+$ NK cells from Bw4$^-$ donors, those that were educated by Bw4$^+$I80N cells had a mean granzyme B MFI of 17,500 ± 2,900 (Fig 6E and F). This was significantly above the mean level (13,500 ± 2,300) of NK cells educated by Bw4$^-$L82R cells (Fig 6E and F).

In combination, these data indicate that in vitro education with Bw4 increases the homeostatic level of intracellular granzyme B, giving a phenotype like that of NK cells educated in vivo. This phenotype was only observed when initial education was followed by further stimulation and a period of rest.

### A synthetic mediator of NK cell education

To test whether viable target cells are required to educate NK cells in vitro, we made synthetic targets displaying both HLA class I and ligands for activating NK cell receptors. First, streptavidin-coated paramagnetic beads were coated with two biotinylated monoclonal antibodies; one specific for CD2, and the other specific for DNAM1. The resulting anti-CD2-DNAM1 beads were then co-cultured with NK cells in medium with high IL-2. This culture enables the beads to bind and extract HLA class I from NK cell surfaces. After 48 h, anti-CD2-DNAM1 beads were isolated from the co-culture with a magnet. Beads were tested for bound HLA class I using FITC-labeled W6/32 antibody and were examined by light and fluorescence microscopy. Negative control beads were not cultured with NK cells.

The beads were distinguished from cells and cellular debris using light microscopy (Fig 7A). No auto-fluorescence was detected on unstained beads in the FITC channel (Fig 7A, first column). Among beads stained with FITC-W6/32, no fluorescence was detected on beads that were not cultured without NK cells (Fig 7A, second column). Anti-CD2-DNAM1 beads that were co-cultured with NK cells stained with fluorescent FITC-W6/32 (Fig 7A, third column). This indicates that HLA class I from NK cells was deposited on the beads during co-culture. The fluorescent signal was not diminished upon repeated washing and centrifugation of the beads (Fig 7A, fourth column). Anti-CD2-DNAM1 beads that have been co-cultured for 48 h with isolated NK cells are subsequently termed as reverse trogocytosis (RT) beads because the process by which they acquire HLA class I is like RT (Carlin et al, 2001).

To determine if one or both activation ligands on the anti-CD2-DNAM1 beads are necessary for RT, we made beads coated with only one of the anti-ligand antibodies. We then co-cultured isolated NK cells with anti-CD2 beads, anti-DNAM1 beads, or anti-CD2-DNAM1 beads in high IL-2 medium. A further control was a culture of anti-CD2-DNAM1 beads without NK cells. After 48 h, we magnetically isolated the beads from their respective co-cultures, stained them with W6/32, and analyzed them by flow cytometry. Anti-CD2 beads had a W6/32 MFI of 1831, which was higher than the 1086 W6/32 MFI of anti-CD2-DNAM1 beads (Fig 7B). RT-anti-DNAM1 beads had a W6/32 MFI of 65, an amount comparable with the MFI of beads cultured without NK cells, which was 30 (Fig 7B). This pattern indicates that the observed RT is mediated solely by the interaction between anti-CD2 antibody and CD2.

### Beads coated with Bw4$^+$HLA-B mediate in vitro NK cell education

Anti-CD2-DNAM1 beads were incubated with NK cells derived from two donors. One donor had two copies of Bw4$^+$*HLA-B* (*HLA-B*51* and *-B*57*), and the other donor lacked Bw4 (*HLA-B*08* and *-B*62*). This provided Bw4$^+$RT and Bw4$^-$RT beads, respectively. The presence of Bw4$^+$HLA-B was confirmed on Bw4$^+$RT beads using a Bw4-specific antibody (Fig S2E).

To determine if beads coated with HLA-B deposited by RT were sufficient to educate NK cells in vitro, isolated NK cells were co-cultured with either the Bw4$^+$RT beads or the Bw4$^-$RT beads for 5 d in medium with high IL-2. On day 5, the medium was replaced, and co-cultures continued for another 2 d. On day 7, NK cells were separated magnetically from beads and then co-cultured with K562 cells. KIR3DL1$^+$NKG2A$^-$ NK cells were then assayed for IFNγ expression.

On average, 45.6% ± 18.5% of KIR3DL1$^+$NKG2A$^-$ NK cells educated by Bw4$^+$RT beads produced IFNγ. This was significantly more than the 38.4% ± 14.9% achieved by the NK cells co-cultured with Bw4$^-$RT beads (Fig 7C). This result shows that Bw4$^+$RT beads display the ligands necessary for some degree of in vitro NK cell education.

## Discussion

This investigation demonstrates that education of mature peripheral KIR3DL1$^+$ NK cells is feasible through in vitro interactions of KIR3DL1

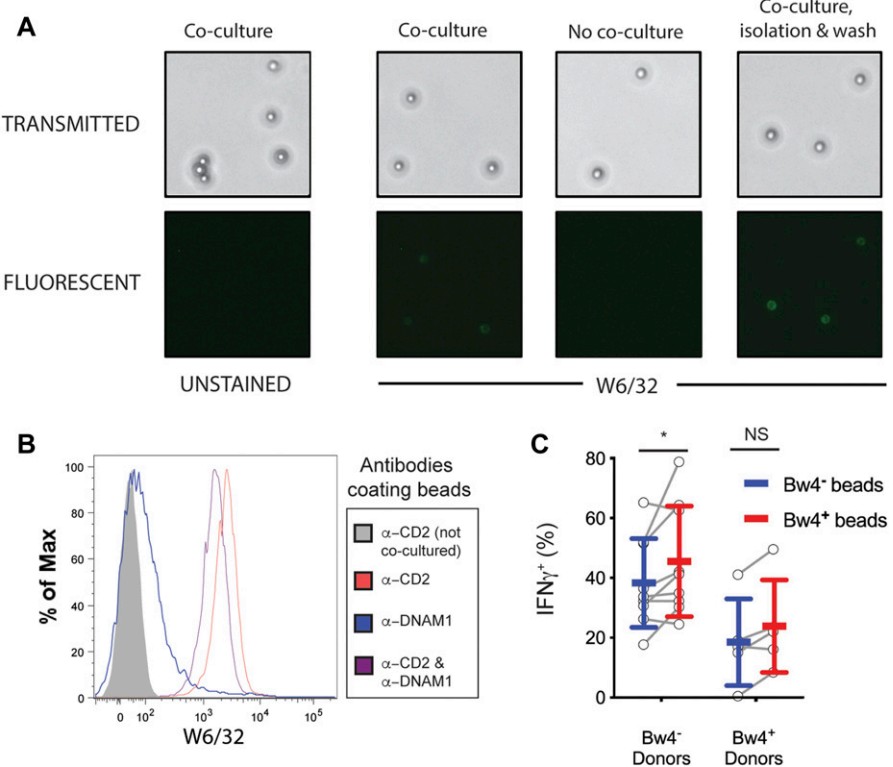

**Figure 7. Synthetic beads displaying Bw4+HLA-B are sufficient to educate KIR3DL1+ NK cells in vitro.** **(A)** Biotinylated antibodies against CD2 and DNAM1 were bound to streptavidin-coated paramagnetic beads and co-cultured with NK cells isolated from PBMC for 2 d in medium with 500 U/ml rhIL-2. The beads were separated with a magnet, stained with anti-HLA class I, and analyzed by light and fluorescent microscopy. **(B)** NK cells were co-cultured for 2 d with paramagnetic beads coated with either anti-DNAM1, anti-CD2, or both anti-DNAM1 and anti-CD2 antibodies. Co-cultures occurred in medium with 500 U/ml rhIL-2. Beads were isolated with a magnet, stained with anti-HLA class I and Alexa Fluor 700–labeled streptavidin, and analyzed by flow cytometry. Shown is the HLA signal of particles in the streptavidin+ gate, which excludes all cells. Representative of at least three replicate experiments. **(C)** Two sets of reverse-trogocytosed beads were constructed by co-culturing beads coated with anti-DNAM1 and anti-CD2 antibodies with NK cells from either Bw4+ or Bw4− donors. This produced beads coated in NK cell-derived membranes displaying Bw4+ or Bw4− HLA-B. Beads were then isolated with a magnet, washed, and co-cultured with NK cells isolated from either Bw4+ or Bw4− donors. Co-cultures occurred in medium with 500 U/ml rhIL-2. After 7 d, NK cells were separated from the beads with a magnet and combined with K562 cells at a 10:1 E:T ratio. 6 h later, the NK cells were stained with antibodies and analyzed by flow cytometry. Shown is combined data comprising nine Bw4− donors and five Bw4+ donors from three replicate experiments. Shown are mean ± SD and the results of a Sidak's multiple comparison tests from a paired two-way ANOVA. NS = not significant. *P < 0.05. Source data are available for this figure.

with Bw4+HLA-B. As predicted by studies of in vivo education (Yawata et al, 2006), in vitro education of KIR3DL1+ NK cells in our model requires the presence of an intact and accessible Bw4 epitope (Carr et al, 2005) and varied in magnitude according to the strength and number of KIR3DL1-HLA-B bonds (Yawata et al, 2006; Kim et al, 2008; Boudreau et al, 2016).

Although HLA-B has a half-life of about 12 h on the cell surface (Yarzabek et al, 2018), blocking HLA-B on BCL training cells at the outset of a 5-d training period abrogated in vitro education. This implies that the signals needed for in vitro education occur early in the training phase. In developing the training protocol, we found that the timing of training was dependent on the training cell line used and that inconsistent results were obtained if NK cells were trained for periods of only 24 or 48 h. These observations, therefore, imply that the signaling necessary for in vitro education occurs at the outset of training and is followed by several days of maturation. That maturation is required for NK cell education is also supported by our observation that increased granzyme B levels, a phenotype associated with in vivo education, are only seen on NK cells from Bw4− donors after 5 d of training and 2 d of rest.

Throughout our study, we intentionally chose HLA-B alleles that bind strongly to KIR3DL1 for use in our Bw4+ training cells or beads. In most conditions, this resulted in improvements in NK cells from Bw4− donors as well as those from Bw4+ donors, which already had achieved some degree of in vivo education through Bw4. The possibility of improving NK cells that have already experienced in vivo education greatly increases the number of human genotypes that

could benefit from immunotherapy based on autologous in vitro NK cell education.

One potential issue with therapeutic applications of in vitro NK cell education is that such education is unlikely to be permanent. Educated mouse NK cells are de-educated when placed in an MHC− mouse (Wu & Raulet, 1997; Joncker et al, 2010), suggesting that in vitro education would also be reversed in human NK cell immunotherapies. Likewise, in our experiments, in vivo education between Bw4+ and Bw4− donors was not evident after training with Bw4− cells. This suggests that in vivo education declines after exposure to Bw4− targets. In contrast, we also report the encouraging observation that in vitro NK cell education mediated by KIR3DL1 is sustained after more than one exposure to HLA− cells, supporting a model in which human NK cell education is plastic, but perhaps more easily gained than lost.

Opinions differ on whether, and under what circumstances, inflammatory cytokines can compensate for a lack of NK cell education (Juelke et al, 2009; Fauriat et al, 2010; Wagner et al, 2017). Although additional rhIL-2 increased the production of IFNγ in our experiments, the effect of education on degranulation declined in medium with rhIL-2 and no other cytokines. Adding rhIL-12 during in vitro NK cell education reversed this situation, allowing education to be detected by degranulation but not IFNγ production alone. These points outline a model in which NK cells educated in homeostatic or mildly stimulatory conditions support cytokine production, whereas those educated in more inflammatory conditions support degranulation.

A potential weakness of our experimental design is that culturing NK cells for several days with cytokines induces a variety of transcriptional and metabolic changes that could confound the detection of NK cell education. Such changes include lysosomal biogenesis (Goodridge et al, 2019) and alterations in metabolism via the mTOR pathway (Marcais et al, 2017; Almutairi et al, 2019). Induction of these cellular programs also likely contributes to donor-specific variation of cell cultures.

Our experiments have not identified the co-factors that are sufficient for NK cell education. However, Bw4+ training consistently improved the missing-self response of NK cells by ~30%, regardless of whether BCL or 721.221 cells were used for training. This observation suggests that, apart from activation receptor ligands and KIR ligands, other co-factors are either ubiquitous or unnecessary for NK cell education. By contrast, KIR3DL1+ NK cells educated in vivo generally outperform uneducated NK cells by more than 30% (Elliott et al, 2010), implying that our training did not achieve the highest degree of NK cell education. As shown by our analysis of in vivo education, one reason for this discrepancy is likely the duration of cell culture.

More robust in vitro education might be achieved if NK cells can be educated in vitro under more homeostatic conditions. However, the extent to which the activation of NK cells is required for in vitro education remains to be defined. It cannot be the case that homeostatic interactions between NK cells count toward education to the same degree as interactions between NK cells and training targets. Otherwise, in vitro education would be unlikely to take place because NK cells outnumbered training cells throughout our experiments. Moreover, training in media with high IL-2 & IL-12 resulted in the most robust signal of in vitro education. A more rigorous study of metabolic and activation signaling during NK cell training will be necessary to define the role each has on the process of in vitro education.

Another potential reason that in vitro education appeared less effective than in vivo education is that our analyses excluded many NK cells that expressed other classes of inhibitory receptors contributing to NK cell education, such as NKG2A and LILRB1 (Sullivan et al, 2008). Additional study using a variety of training cell lines and conditions will be necessary to optimize in vitro education, and will likely include education through multiple classes of inhibitory receptors.

One mechanistic interpretation of our data is that NK cells receiving inhibitory signaling during the training phase were able to conserve functionality for the testing phase. In this scenario, KIR3DL1+ NK cells in the Bw4− training condition became more activated during training than those in the Bw4+ condition, thus depleting cellular resources. Then, in the testing phase, NK cells trained with Bw4+ had more resources available to respond to missing-self. Such an interpretation fits with the disarming model of NK cell education, in which a lack of inhibitory signaling erodes NK cell functionality over time (Raulet & Vance, 2006).

A possible contradiction of this model is that NK cells cultured alone should conserve the most resources. Yet, untrained NK cells in our experiments did not outperform NK cells trained by Bw4. However, because of the extensive reprogramming induced by target interaction, it is unrealistic to make direct comparisons between trained and untrained NK cells. Therefore, the disarming model remains one reasonable interpretation of our data.

The CD58−CD2 bond has low affinity (Davis et al, 1998) but is required for the formation of membrane nanotubes between NK cells and targets (Comerci et al, 2012). This may explain why NK cell membranes are disrupted and left behind after contact with beads displaying anti-CD2. RT beads introduce a novel tool, not only for the study of NK cell education but also for studies involving HLA presentation. For instance, by first infecting NK cells and then co-culturing them with beads, RT beads displaying HLA with pathogen-specific epitopes could be easily isolated.

# Materials and Methods

### Target cell lines

B lymphoblastoid cell lines (BCLs) were generated by combining $10^7$ PBMCs with the B95.8 strain of EBV, as described previously (Neitzel, 1986). BCLs were cultured in RPMI-1640 medium (Corning) containing 10% FBS (Corning), 2 mM L-glutamine (Thermo Fisher Scientific), and 100 U/ml of penicillin and streptomycin (Thermo Fisher Scientific). This medium is hereafter called RPMI10%-C.

Both BCL cell lines were homozygous for the C1 epitope of HLA-C, and lack the A3/11 epitope of HLA-A. Thus, both BCLs express the C1 epitope recognized by KIR2DL2 and KIR2DL3. In contrast, one BCL (Bw4+BCL) expressed the Bw4 epitope recognized by KIR3DL1, whereas the other BCL (Bw4−BCL) does not. Bw4+BCL expressed HLA-B*51 and B*57, whereas Bw4−BCL expressed HLA-B*08 and B*62.

721.221 transfectant cell lines were generated and maintained as described previously (Sanjanwala et al, 2008). In summary, plasmid expression vector pEF6-V5-His containing either *B*5801*, *B*0801*, or mutant *B*5101* cDNA was transfected into 221 cells by electroporation using Gene Pulser (Bio-Rad Laboratories). Afterward, 221 cells were cultured and maintained under selection in RPMI10% with 5 μg/ml blasticidin (Invitrogen).

Site-specific mutation of *HLA-B*5101* was performed as previously described (Ho et al, 1989; Sanjanwala et al, 2008). In summary, two oligonucleotide primers (Stanford PAN Facility) were designed to encode the desired mutation and to bind to the area surrounding the mutation site, one each for the 5′ and 3′ DNA strands. Each of those primers was used in a separate PCR to amplify either the cDNA sequence upstream or downstream from the mutation site. The resulting amplicons were then purified and combined in a second PCR reaction, thereby forming the full-length mutant cDNA. In one mutant, residue 82 of B*51:01 was mutated to encode arginine instead of leucine (Bw4−221-B*51-L82R). This mutation disrupts the Bw4 epitope, thereby preventing the binding of KIR3DL1, as described previously (Sanjanwala et al, 2008). In another mutant, residue 80 of B*51:01 was mutated to encode asparagine instead of isoleucine (Bw4+221-B*51-I80N). This alteration does not disrupt the Bw4 epitope.

### Blood acquisition and processing

Leukoreduction and separation chambers obtained from healthy, CMV-free donors were purchased from the Stanford Blood Center. PBMCs from the leukoreduction and separation chambers were isolated on a Ficoll-Paque gradient (GE Healthcare) and then suspended

in FBS (Corning) containing 10% DMSO (EMD Millipore) at $10^7$/ml. Aliquots were frozen at −80°C for several days using a Mr. Frosty device (Thermo Fisher Scientific) and then stored in liquid nitrogen.

Before experiments, frozen aliquots of PBMCs were thawed at 37°C in a water bath and suspended in 10 ml of 37°C RPMI10%-C. The cells were then spun at 300$g$ for 10 min, resuspended in 2 ml of RPMI10%-C with 100 U/ml recombinant human IL-2 (rhIL-2), and transferred to 12-well plates at $10^7$ cells per well. The cells were kept in a 37°C incubator with 5% $CO_2$ for >12 h before any further manipulation. The IL-2 was obtained from Dr. Maurice Gately (Hoffmann-La Roche Inc.), through the National Institutes of Health AIDS Reagent Program, Division of AIDS, National Institute of Allergy and Infectious Diseases, National Institutes of Health.

### HLA and KIR genotyping

*HLA* and *KIR* were genotyped as described previously (Norman et al, 2016). In summary, DNA was extracted from whole blood using the QIAamp DNA Blood Mini Kit (QIAGEN) following the manufacturer's instructions. Genomic DNA fragments representing complete *HLA-A*, *HLA-B*, *HLA-C*, and *KIR* genes were isolated using oligonucleotide probes and sequenced using an Illumina MiSeq machine with v3 technology (Illumina Inc.), as previously described (Norman et al, 2016).

### Donor selection

Bw4⁻ individuals were defined as having no Bw4 epitope on either HLA-A or HLA-B. Bw4⁺ donors were defined by the presence of Bw4 on HLA-B. Donors were selected to be homozygous for the C1 epitope of HLA-C.

### NK isolation

NK cells were isolated from PBMCs using the Untouched NK Isolation Kit with LS columns (Miltenyi Biotec), as described previously (Pugh et al, 2018). In this procedure, all PBMCs except NK cells were bound to paramagnetic beads by antibodies. This mixture was then passed through a column in the presence of a magnet, which trapped all PBMCs, except NK cells, in the column. The NK cell–enriched flow-through was then washed and suspended in RPMI10%-C.

### Etoposide treatment of training lines

Training lines in all experiments were subjected to etoposide treatment before co-culture with NK cells. ~$10^6$ BCL or 221 cells in RPMI10%-C were transferred to a six-well plate at 10 ml/well. VP-16 was added to the media. The final concentration of VP16 etoposide was determined empirically for each training line to halt division, generally 125–250 mM. The cells were then incubated at 37°C in 5% $CO_2$ for 4 h. The cells were then spun at 300$g$ and washed with 25 ml RPMI10%-C three times before being used in NK cell co-cultures.

### In vitro education using B cell lines

200,000 NK cells from each donor were co-cultured with VP16-treated BCL target cells at an 8:3 E:T ratio in flat-bottom 96-well plates. The cells were co-cultured in X-vivo 15 medium (Lonza) with 10% FBS (10% X-vivo). The medium included either 100 U/ml rhIL-2, 500 U/ml rhIL-2, or 500 U/ml rhIL-2 and 50 ng/ml rhIL-12 (eBioscience/Thermo Fisher Scientific). The cells were co-cultured at 37°C in 5% $CO_2$ for 5 d, without disturbance. NK cells were then assayed for their response to K562 cells.

### In vitro education using 221 transfectant lines

200,000 isolated NK cells were co-cultured with VP16-treated 221 cells at a 1:2 T:E ratio. The cells were co-cultured in 96-well flat-bottom plates in 10% X-vivo medium with 500 U/ml rhIL-2 at 37°C in 5% $CO_2$. The medium was checked daily for discoloration and changed as needed, at most every 24 h. To change the medium, co-cultured cells were transferred to round-bottom 96-well plates, pelleted at ~700$g$, resuspended in fresh medium, and then returned to flat-bottom plates. After 3 d, the NK cells were assayed for their response to K562 cells.

### Activation beads and RT

Activation beads were constructed using the NK Cell Expansion Kit, as directed by the manufacturer (Miltenyi Biotec). In summary, biotinylated antibodies were bound to paramagnetic beads coated with streptavidin. Antibodies comprised anti-CD2 (Miltenyi Biotec) and anti-DNAM1 (Miltenyi Biotec), either in combination or alone.

RT was accomplished by combining activation beads with 200,000 isolated NK cells at a ratio of 1:2 in 96-well flat-bottom plates in 200 $\mu$l 10% X-vivo with 500 U/ml rhIL-2. 48 h later, each co-culture was agitated by repeated pipetting and then transferred to a 1.2-ml microtiter tube (Thermo Fisher Scientific) containing 100 $\mu$l of X-vivo 15 medium. To harvest RT beads, the microtiter tube was suspended in the open groove of a horizontally positioned MidiMACS separator magnet (Miltenyi Biotec). After 5 min, the entire volume of the medium was gently extracted from the bottom of the microtiter tube, leaving the beads held by the magnet on the inner sides of the tube. The beads were immediately resuspended by removing them from the magnetic field and adding RPMI-1640. This magnetic isolation process was repeated to further purify the bead fraction. Last, suspensions of beads were transferred to a 1.5-ml tube and stored at 4°C for several days before experimentation.

### In vitro education using RT-beads

Beads were constructed, subjected to RT, and isolated as described above. 200,000 NK cells were combined with RT beads in a flat-bottom 96-well plate at a 5:1 cell : bead ratio in 10% X-vivo medium with 500 U/ml rhIL-2. Co-cultures were incubated undisturbed for 5 d at 37°C in 5% $CO_2$. On day 5, co-cultures were transferred to a 96-well round-bottom plate, pelleted by centrifugation at 300$g$, suspended in fresh medium and returned to flat-bottom plates. Co-cultures were then incubated for 2 more days. On day 7, co-cultures were transferred to a microtiter tube and NK cells were separated from beads using a MidiMACS magnet, as described above. Isolated NK cells were then assayed for their response to K562 cells.

## Functional assays

After in vitro education, NK cells were transferred to a round-bottom 96-well plate, pelleted by centrifugation at ~700$g$, and suspended in 100 $\mu$l of 10% X-vivo media with 500 U/ml rhIL-2 and 5 $\mu$g/ml Brefeldin A (Sigma-Aldrich). For degranulation assays, each well contained 2 $\mu$l of anti-CD107a antibody (eBioH4A3; eBioscience/Thermo Fisher Scientific). Target cells were then added to each well at a 1:10 T:E ratio. Co-cultures were incubated for 6 h at 37°C in 5% $CO_2$. Co-cultures were then pelleted by centrifugation at 4°C, washed twice with ice-cold magnetic assisted cell separation (MACS) buffer, and stained with antibodies to detect cell surface proteins.

## F(ab')$_2$ production

W6/32 antibody was purified from the spent medium of cultured W6/32 hybridoma cells as described previously (Parham et al, 1979). F(ab')$_2$ were prepared from W6/32 antibody using the Pierce F(ab')$^2$ Preparation Kit (Thermo Fisher Scientific), following the manufacturer's protocol. In summary, 1-mg batches of W6/32 antibody were added to spin columns containing equilibrated immobilized pepsin. Fc was then digested by pepsin for 1 h at 37°C. Undigested IgG was removed by binding to Protein A. Batches of purified F(ab')$_2$ were tested for their ability to block whole W6/32 antibody (Fig S2A).

## CMV infection

Human CMV strain VHL/E was a kind gift from James Waldman (Waldman et al, 1989). MRC-5 cells were acquired from American Type Culture Collection and cultured in Eagles Minimum Essential Medium (EMEM) (Gibco) with 10% FBS (Corning). MRC-5 cells were infected overnight in serum-free EMEM, then cultured in 10% EMEM for 4 d postinfection before being assayed for Bw4, or used as testing targets for in vitro education assays.

## Antibodies

Seven fluorescently conjugated monoclonal antibodies specific for cell surface proteins were used to analyze NK cells. These comprised anti-CD19 (HIB19; eBioscience/Thermo Fisher Scientific), anti-CD-107a (eBioH4A3; eBioscience/Thermo Fisher Scientific), anti-KIR3DL1 (DX9; BioLegend), anti-NKG2A (REA110; Miltenyi Biotec), anti-LILRB1 (GHI/75; BioLegend), anti-panKIR2D (NKVFS1; Miltneyi), and anti-DNAM1 (11A8; BioLegend). Four fluorescently conjugated monoclonal antibodies were used to detect the following intracellular markers: anti-IFN$\gamma$ (45.B3; BioLegend), anti-BrdU (51-23619L; BD Pharmingen), anti-granzyme B (QA16A02; BioLegend), and anti-granzyme B (GB11; BD Bioscience).

## Cell staining, fixation, and flow cytometry

Cultures containing NK cells were stained in U-bottom, 96-well plates with antibodies specific for cell surface proteins, as described previously (Pugh et al, 2018). The staining concentration of each antibody was determined empirically, ranging from 1 to 3 $\mu$l per test. The cells were stained in a total volume of 50 $\mu$l PBS with 0.5% bovine serum albumin (Sigma-Aldrich) and 2 mM EDTA (MACS buffer). The cells were stained in the dark at 4°C for a period of between 0.5 and 12 h. The stained cells were then washed with 200 $\mu$l PBS and pelleted by centrifugation at ~700$g$ for 3 min at 4°C. This washing step was repeated twice. During subsequent steps, the reagents and plates were kept on ice.

To identify and exclude dead cells from analysis, cultures were then stained with a 500-fold dilution of Live/Dead Yellow (Thermo Fisher Scientific) in PBS. The cells were stained in a total volume of 50 $\mu$l for 30 min at 4°C. The cells were washed twice with MACS buffer and then treated with 100 $\mu$l of fixative solution (BD Cytofix; BD Biosciences). After 10 min of incubation at 4°C, 100 $\mu$l of perm buffer (1 × BD Perm; BD Biosciences) were added to each well, to permeabilize the cells. The cells were then washed twice with 200 $\mu$l of perm buffer. To ensure the permeability of intracellular compartments, the cells were incubated with 200 $\mu$l of perm buffer containing 10% DMSO, at 4°C for 20 min, followed by three more washes in perm buffer. The cells were then treated again with 100 $\mu$l fixative solution to ensure the fixation of intracellular compartments, followed by two more washes with perm buffer. For BrDU detection, DNA was digested by incubating cells with 30 $\mu$g/ml DNase I (Sigma-Aldrich) in perm buffer at room temperature in the dark for 30 min. To detect BrDU or intracellular proteins, the cells were stained with antibodies in 50 $\mu$l of perm buffer for 30 min at 4°C. The cells were washed twice with perm buffer and once with MACS buffer before analysis.

The cells were analyzed using an LSR II flow cytometer (BD Biosciences) at the Stanford Shared FACS Facility. Sphero midrange Rainbow Fluorescent beads (Spherotech) were used to calibrate and verify the performance of all channels before each data collection session. The same cytometer was used when comparing samples from the same experiment collected on different days. Data points appearing in paired comparisons were collected on the same cytometer in the same day without interruption. Data were analyzed using FlowJo Ver. 9.7.6 (FlowJo, LLC). Concatenation was performed using Flowjo Ver. 10.6.1.

## B&EF calculation

Binding scores were generated by color quantifying the heat map dataset of (Boudreau et al, 2016) using Adobe Photoshop CS6 Version 16 (Adobe Inc.). Binding quantities were normalized on a continuous scale from 1 to 10. Donors for whom all KIR-HLA-B combinations were represented in the binding data were selected. Each of the four possible HLA-B-KIR bonds when training with the Bw4[+] cell line was calculated for each donor. Scores for the four bonds were then averaged, resulting in a single overall binding score for each donor. The overall binding score for each donor was then multiplied by 1,000 and multiplied by the MFI of KIR3DL1 for that donor. The KIR3DL1 MFI was recorded after NK cell isolation but before training or any co-culture.

## Microscopy

Microscope images were captured using an EVOS FL microscope (Life Technologies), using a 20× LPlanFL lens. Fluorescent images were captured using an EVOS LED Cube GFP (F1013-1410-314) and EVOS software (revision 21773). Paired transmission and fluorescent images were taken with the same positioning and cropped to display the same cell grouping using Adobe Photoshop CS6 version 13 (Adobe Inc.).

## Statistics

Data were assessed for statistical significance using Prism Ver. 6 (GraphPad Software). Generally, paired samples from the same donor were compared with a paired two-way ANOVA.

# Supplementary Information

# Acknowledgements

We thank Amir Horowitz, Emily Wroblewski, Hugo Hilton, and Katherine Walwyn-Brown for technical and scientific input. We also thank the Stanford Shared FACS Facility for flow cytometry and sorting support. This work was supported by a grant from the National Institutes of Health, AI017892, to P Parham.

## Author Contributions

J Pugh: conceptualization, data curation, formal analysis, investigation, project administration, and writing—original draft, review, and editing.
N Nemat-Gorgani: data curation and formal analysis.
Z Djaoud: investigation.
LA Guethlein: data curation and formal analysis.
PJ Norman: data curation and formal analysis.
P Parham: supervision, project administration, and writing—original draft, review, and editing.

## Conflict of Interest Statement

The authors declare that they have no conflict of interest.

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
