## [Reviewer comments · Life Science Alliance]

Life Science Alliance

In Vitro Education of Human Natural Killer Cells by KIR3DL1.

Jason Pugh, Neda Nemat-Gorgani, Zakia Djaoud, Lisbeth Guethlein, Paul Norman, and Peter Parham

DOI: <https://doi.org/10.26508/lsa.201900434>

Corresponding author(s): Peter Parham, Stanford University

Review Timeline:

Submission Date:	2019-05-20
Editorial Decision:	2019-06-13
Revision Received:	2019-10-06
Editorial Decision:	2019-10-22
Revision Received:	2019-11-01
Accepted:	2019-11-04

Scientific Editor: Andrea Leibfried

Transaction Report:

June 13, 2019

Re: Life Science Alliance manuscript #LSA-2019-00434-T

Prof. Peter Parham
Stanford University
Stanford University Medical Ctr.
School of Medicine
Dept. of Cell Biology
Stanford, Sherman Fairchild Bldg. D157 USA-Stanford, CA 94305

Dear Dr. Parham,

Thank you for submitting your manuscript entitled "In Vitro Education of Human Natural Killer Cells" to Life Science Alliance. The manuscript was assessed by expert reviewers, whose comments are appended to this letter.

As you will see, all three reviewers appreciate your data but think that the proposed in vitro education needs to get further corroborated and tested to better support your conclusions. We would thus like to invite you to submit a revised manuscript to us. While the requested mechanistic insight (reviewer #1) is not mandatorily needed for publication here, all other concerns of the reviewers should get addressed. Importantly, the functionality of the educated NK cells should get tested and the education further corroborated (reviewer #1 and #2) and the experimental set-up should get re-visited (reviewer #2). An alternative explanation for the observed effects needs to get considered as well (reviewer #2) and the data presentation should get revised as outlined by reviewer #3.

Thank you for this interesting contribution to Life Science Alliance. We are looking forward to receiving your revised manuscript.

Sincerely,

B. MANUSCRIPT ORGANIZATION AND FORMATTING:

Reviewer #1 (Comments to the Authors (Required)):

In this work Pugh et al examine whether the phenomenon of NK cell education through interaction with inhibitory KIR receptors and specific HLA epitopes can be mimicked in vitro. The authors use a human system and focus on NK cells expressing the inhibitory receptor KIR3DL1. Based on the data presented, the authors conclude that NK cell education can indeed be accomplished in vitro. The work is interesting, and the described method, opens up for a whole range of questions which can now be addressed. However, I think the work is under-developed regarding the general application of the phenomenon in NK cell biology as well as underlying mechanism.

1. The in vitro-educated NK cells should be functionally tested in a disease-relevant setting, e.g. virus infections.
2. The work appears very descriptive for me. The authors should provide some form of mechanistic explanation for the observation.
3. To make the general statement in the title, the authors should demonstrate the phenomenon for more than one NK cell receptor.

Reviewer #2 (Comments to the Authors (Required)):

In this study the authors have examined the possibility to educate NK cells through an in vitro education session (training) using a co-culture system with feeder cells expressing self /non-self ligands. The data suggest that training on self MHC leads to increased effector responses, in particular for IFN-gamma, during the test phase. This effect is modulated by IL-12 and MHC class I blockade. A nice aspect of the paper is the possibility to use synthetic ligands coupled to beads to tune NK cell function. Overall, the possibility to tune NK cell function in vitro to enhance missing-self responses is of potential importance for NK-cell based cancer immunotherapy. The biggest concern is the resolution (or variation?) of the experimental model given that it is hardly capable of reading out in vivo education. Education is a very robust phenomenon, but there is a huge donor variation and the functional read-outs used to probe education are heavily influenced by cytokines and other experimental conditions. Therefore, it is essential to perform a high number of experiments using many donors in order to filter out real effects from the background variation. If corroborated, the study has potential to further our understanding on the cell-cell interactions that tune effector responses in NK cells.

- 1) The effect in the different experimental series performed throughout the study largely point in the same direction and provide support for the claims made. However, it is a bit frustrating that the effect is not robustly shown in Figure 2, which is the basis for the remaining experiments. In particular the CD107a experiments in K562 seems suboptimal? It is worrying that an experimental system that seeks to explore induced education, cannot detect differences in education between KIR3DL1+ NK cells in Bw4+ versus Bw4- donors. If this is a consequence of 5 days in IL-2 or other experimental conditions remains unclear. This must be revisited. Degranulation levels at 4% is much lower than expected with K562 cells? The results in Figure 2D after IL-12 (and 3B) are more convincing although it remains a concern that natural in vivo education appears gone also in this setting?
- 2) Ad1: The data without training for IFN shown in Supp Figure 1B-C seems to suggest that the education impact is partly reduced by feeders (and possibly better retained under the no training

conditions)? Since CD107a was included in the assays it seems odd not to show those results?

3) As alluded to in the general comment, the effects noted in this experimental model are rather small, which may possibly be a consequence of the length of stimulation with IL-2 used during the training phase. Most currently used read-outs on NK cell education are based on ex vivo or in vivo analysis of resting NK cells or short term activated NK cells. The hyporesponsiveness of uneducated NK cells is partly reversed by exposure to cytokines (this study and previous studies) that affect the levels of effector molecules (granzyme B), lysosomal biogenesis (Malmberg Lab NCOMMS 2019), mTOR activation and metabolism (Walzer Lab, ELIFE 2017 and Lee Lab, JACIE 2018), with bearing on NK cell effector function. I realize that it is not realistic to redo all the experiments using shorter training periods (for example 48 hours) but it would be very exciting to see the outcome of experiments performed before major effects off metabolic and transcriptional reprogramming have taken place and before onset of cell proliferation. The authors may want to comment on whether it may be feasible to study the impact of training with self ligands under more homeostatic conditions and possibly discuss this weakness.

4) In the results section, the authors provide evidence demonstrating that the observed effects are not "simply" based on more activation of NK cells exposed to targets expressing self ligands during the training phase (Fig 2F). While I see the importance of excluding effects on cell survival and proliferation, the point dealing with activation is more complex, since it may in fact be the opposite effect (disarming) we are seeing during the training phase? Indeed, somewhat confusingly, in the discussion the authors focus on this latter effect and refute disarming as an explanation for the observed in vitro education.

One would expect that cells exposed to self ligands should be less activated during the training phase. In fact, this is a scenario we anticipate during in vivo education (although at much lower cytokine concentrations). I believe the data presented are compatible with the disarming model of NK cell education. Priming under self inhibition would protect from disarming. Such potential protection from disarming would have to be monitored during the training phase, corresponding to the no test (no target) conditions at various time-points after training. Although I agree with the authors that one would possibly expect the training effect to be lost after exposure to K562 in the test phase, I do not think there is sufficient evidence to totally refute disarming as an underlying principle behind the relative increase in function in cells that are inhibited during the priming phase. One possible reason for such a preserved effect could be more long-term changes in metabolism in cells that are primed in IL-2 under inhibition versus those are not.

The argument made about the superior stimulatory effect of any feeder (carrying self and non-self ligands) versus cytokine alone is not an argument against disarming since feeders induce a much stronger proliferative response and the two conditions are not possible to compare. Cytokine stimulation alone has been shown to dampen NK cell functionality through effects on metabolism.

I guess what I am trying to say that manipulating/controlling such disarming events in vitro would be of no less importance (and would still be referred to as education) than any other unknown mechanism, operating during the in vitro training. Please consider softening the strong statements against this alternative (eg disarming) explanation.

5) It is interesting to see that the authors can detect differences in granzyme B levels, in keeping with the study of Goodridge et al. However, effects of priming are important to consider in this context since IL-2 leads to a 100-1000 fold increase in granzyme B levels compared to baseline. Thus, granzyme B is likely a poor proxy of education under conditions of cytokine priming, in

particular in the absence of a rest period. Would it be helpful to calculate the ratio of granzyme B expression between self versus non-self KIR+ NK cells to monitor how this change after 5 days in IL-2 and how is this changed by the training? Should be possible to extract from the present data and compared to baseline levels without priming?

6) Typo in figure legends Figure 2. Something is missing. "donors"?

7) Figure 3E/4B: Replace no testing with no target?

Reviewer #3 (Comments to the Authors (Required)):

NK cell education has been shown to affect NK cell activity in response to stimulation. This is an important area in the NK field as the molecular underpinnings of education remain poorly defined. The current work thus sought to establish an in vitro system to educate NK cells.

Overall, the study is important and relevant, and its findings are interesting and compelling. As shown, the newly developed in vitro assay showed that 3DL1+ NK cells were positively affected by co-culture with target cells or synthetic beads presenting its specific HLA-B ligand. The data further showed the effect is dependent on the KIR receptor, as well as its ligand. Though the magnitude of the in vitro education effect was rather small in each experiment, it was highly consistent and significant. Moreover, the study used several strategies to verify the main findings which adds to its overall conclusions. Developing such an in vitro approach to license NK cells should help to elucidate the underlying mechanisms related to NK education.

One concern with the manuscript and data shown, however, is its assessment of whether 3DL1+ NK cells can be educated on HLA- targets. Since this experimental setup represented a missing-self scenario, and because the effector NK cells were treated with high dose IL2, it may not be a practical way to evaluate a change in educational status since NK cells should have been activated under these conditions. Another limitation has to do with whether different E:T ratios matter in the case of exposing KIR+ NK cells to HLA- targets during training, which might have impacted the outcome. This is a reasonable question since its been shown in immune competent mouse models that NK cell tolerance to self-MHC I-deficient tumor target cells is dependent on the number of tumor cell targets injected into the host. This question should be carefully studied, though not critical for the main findings shown in the current work. Several points requiring attention are detailed below:

Main Points

1. The results section is tedious and should be shortened considerably. One suggestion is to remove text from the results section which duplicates what is in the methods section, or figure legends.
2. Stats for Bw4+ donors should be included in results shown since differences are discussed. In several cases, data obtained for Bw4+ donors is not even presented.
3. Since it's challenging to compare MFI values in different experiments due to day to day variations in sample processing and cytometer performance, especially for cytometers using PMTs, it would help to clarify (Methods section) that the same Fortessa cytometer was used in both experiments with similar cytometer performance.
4. Histograms are useful for presenting and comparing MFI values for two populations and should be included along with representative gates for all positive fractions. As these data naturally have a large amount of variation, it may help to concatenate whatever number of down sampled events from the positive GZMB fraction for each individual sample within an experiment to generate a

representative graph.

5. Presentation of the details of an approach or experimental setup in some cases is unclear. As one example, development of a synthetic product to educate NK cells is first described using anti-CD2-DNAM1 beads. Quickly this changed to use of anti-CD2-NKp46 beads for most of the experiment for some unknown reason. Why? And then data in Fig 5 indicate anti-CD2 is sufficient to trap HLA molecules. So, why were further studies then performed using anti-CD2-DNAM? This is confusing, and not explained. Related to this, Supp Fig 3c and its legend are in disagreement about what type of beads were actually used in the experiment.

Minor

1. The anti-KIR Ab therapy discussed on p5 has failed in clinical trials.

2. It may help to include discussion of the reasoning for chosen time points, and whether other training time intervals were considered or tested in developing this NK cell education assay.

3. The B&EF equation should be included in the methods section and then briefly mentioned in the results.

4. Box and whisker plots with SD or SEM may aid interpretation of the data. Additionally, the connector lines for training data in 2 different conditions shown in multiple figures are barely visible.

5. Why show Fig 3E stats comparing what happened for Bw4- donor NK cell training to what happened for Bw4+ donor NK cell training?

6. The use of a table to present data (especially for ratios) described in the results section may help to efficiently present and describe the main findings.

7. No CD107a expression data is shown in supp Fig 1 so this mention should be removed from the legend.

8. Discussion text on P27 describing the conservation of potency and a theoretical activation disadvantage is almost impenetrable.

9. Manuscript should be carefully proofread for typos. Some are listed here, but there are others.

- P7 - rhIL2

- *The use of 500IL2+IL12+Bw4+BCL is odd since the cells are not IL2+ or IL12+. It would seem simpler to refer to these cells as cytokine-induced or IL2/IL12-induced.*

- *Supp Figure 1 legend (L.2) - Or[g]ange*

- *Supp Figure 2 legend (L.8) - Bromodoxyuridine*

- *Figure 2A, right panel - rhIL-2[n]*

In our responses below the reviewer's points are given in blue. Our responses follow in black text, with quotes from the revised manuscript indented. For each passage of the revised manuscript that is quoted, we reference the line numbers comprising the quotation in red.

Reviewer #1:
Major points:

Point 1. The *in vitro*-educated NK cells should be functionally tested in a disease-relevant setting, e.g. virus infections.

We thank the reviewer for this important suggestion. We have now included data showing that NK cells educated *in vitro* also demonstrate enhanced functions against the MRC-5 fibroblast cell line when infected with human cytomegalovirus. This data is now presented in Fig. 4E-4F and in the following section of the results:

“Following infection with CMV, MRC-5 cells downregulated their surface expression of HLA-B, as measured by an antibody targeting the Bw4 epitope of HLA-B (Fig. 4E). Among KIR3DL1⁺ NK cells from Bw4⁻ donors, 13.2±12.1% of those trained with the Bw4⁺BCL responded to CMV-infected MRC-5 cells by degranulation and production of IFN γ (Fig. 4F). This was significantly more cells than the 6.5±6.9% degranulation and IFN γ production achieved by KIR3DL1⁺ NK cells trained with the Bw4⁻BCL. KIR3DL1⁺ NK cells from Bw4⁺ donors showed no improvement in their responses to CMV-infected MRC-5 cells due to training with Bw4⁺BCL (Fig 4F). We conclude that *in vitro* education through KIR3DL1⁺ improves the response to missing self, induced by CMV infection.”

(lines 273-282)

Point 2. The work appears very descriptive for me. The authors should provide some form of mechanistic explanation for the observation.

We share the reviewer's desire to define a mechanism through which NK cells are educated. We plan to address the mechanism of NK cell education in future studies using the protocols we have developed here. However, such a mechanism is beyond the scope of this work. At present, we must reference the editor's decision that the lack of a definitive mechanism for NK cell education in this manuscript should not preclude it from publication.

Though not addressed experimentally, mechanism is now discussed in the manuscript with reference to existing models of NK cell education, as follows:

“One mechanistic interpretation of our data is that NK cells receiving inhibitory signaling during the training phase were able to conserve functionality for the testing phase. In this scenario, KIR3DL1⁺ NK cells in the Bw4⁻ training condition became more activated during training than those in the Bw4⁺ condition, thus depleting cellular resources. Then, in the testing phase, NK cells trained with Bw4⁺ had more resources available to respond to missing-self. Such an interpretation fits with the disarming model of NK cell education, in which a lack of inhibitory signaling erodes NK cell functionality over time (Raulet and Vance, 2006).”

(lines 582-589)

Point 3. To make the general statement in the title, the authors should demonstrate the phenomenon for more than one NK cell receptor.

Showing in vitro education by another inhibitory KIR is an important step that we are planning to pursue in future research. However, considering that it took us several years to develop the cohort and protocols to educate NK cells in vitro by KIR3DL1 binding, we felt that it would be impossible to satisfy this point with new data in the 3-month review period. We therefore must refer to the editorial decision provided us in an email from Andrea Leibfried on 6/24/19, in which she agrees that this point can be satisfied by limiting the breadth of our claims both in the body and title of the manuscript.

We have therefore changed the title to be specific to education by KIR3DL1:

“In Vitro Education of Human Natural Killer Cells by KIR3DL1”

(line 1)

Similarly, we have limited our claims of NK cell education to be specific for KIR3DL1 in all sections of the manuscript. This includes in the introduction:

“In this study, for the first time to our knowledge, we provide evidence that mature human peripheral NK cells expressing KIR3DL1 can be educated in vitro.”

(lines 104-106)

Throughout the Results section:

“General approach to in vitro NK cell education by KIR3DL1.”

(line 111)

“In vitro education of **KIR3DL1⁺** NK cells by a B cell line expressing Bw4⁺ HLA-B”

(line 318)

And, in the Discussion section:

“This investigation demonstrates that education of mature peripheral **KIR3DL1⁺** NK cells is feasible through in vitro interactions of **KIR3DL1** with Bw4⁺ HLA-B.”

(lines 499-500)

Reviewer #2:
Major Points:

1) The effect in the different experimental series performed throughout the study largely point in the same direction and provide support for the claims made. However, it is a bit frustrating that the effect is not robustly shown in Figure 2, which is the basis for the remaining experiments. In particular the CD107a experiments in K562 seems suboptimal? It is worrying that an experimental system that seeks to explore induced education, cannot detect differences in education between KIR3DL1+ NK cells in Bw4+ versus Bw4- donors. If this is a consequence of 5 days in IL-2 or other experimental conditions remains unclear. This must be revisited. Degranulation levels at 4% is much lower than expected with K562 cells? The results in Figure 2D after IL-12 (and 3B) are more convincing although it remains a concern that natural in vivo education appears gone also in this setting?

Reviewer #2 raises important and logical concerns in this point that we have addressed in several ways.

Firstly, we agree that the effect of in vivo education is underrepresented in our manuscript. We have therefore added new data, now comprising Figure 2, that highlights in vivo education in our cohort. This additional figure reinforces our findings for in vitro education, in that the effects of both in vivo and in vitro education on degranulation diminish after lengthy cell culture in IL-2. This new data also allowed us to directly address the concern that in vivo education is no longer apparent after lengthy cell cultures in the discussion:

“Our experiments have not identified the co-factors that are sufficient for NK cell education. However, Bw4⁺ training consistently improved the missing-self response of NK cells by ~30%, regardless of whether BCL or 721.221 cells were used for training. This observation suggests that, apart from activation receptor ligands and KIR ligands, other co-factors are either ubiquitous or unnecessary for NK cell education. By contrast, KIR3DL1⁺ NK cells educated in vivo generally outperform uneducated NK cells by more than 30% (Elliott et al., 2010), implying that our experiments did not achieve the highest degree of NK cell education. As shown by our analysis of in vivo education by KIR3DL1, one reason for this discrepancy is likely the duration of cell culture.”

(lines 554-562)

Secondly, we have addressed the reviewer's frustration at the robustness of the data presented by expanding our cohort of donors. The manuscript now also includes the combined results of experiments performed on different cohorts (Table 1), which reinforces our statistical comparisons in key experiments.

Moreover, we shared this reviewer's concern that the CD107a levels we initially reported were low compared to other studies. By re-analyzing our data, it became apparent that we had originally made a mistake in gating. In our prior version of the manuscript, we derived the cut-off for CD107a⁺ cells by applying a common gate to all donors such that *all* unstimulated control samples had 0% CD107a⁺ cells. An unusually activated sample among our controls caused this cutoff to occur at a fluorescence intensity of approximately 5000, which was far too high a threshold. We have applied new gating

that is more in line with the average CD107a level present in unstimulated NK cells from all donors. Happily, when this more realistic gating is applied to our data (Fig. 3B), degranulation frequencies average at approximately 24%. Yet, the relationship between training groups still supports our original point regarding education and degranulation.

2) Ad1: The data without training for IFN shown in Supp Figure 1B-C seems to suggest that the education impact is partly reduced by feeders (and possibly better retained under the no training conditions)? Since CD107a was included in the assays it seems odd not to show those results?

These results are now shown fully in Figure 2B.

3) As alluded to in the general comment, the effects noted in this experimental model are rather small, which may possibly be a consequence of the length of stimulation with IL-2 used during the training phase. Most currently used read-outs on NK cell education are based on ex vivo or in vivo analysis of resting NK cells or short term activated NK cells. The hyporesponsiveness of uneducated NK cells is partly reversed by exposure to cytokines (this study and previous studies) that affect the levels of effector molecules (granzyme B), lysosomal biogenesis (Malmberg Lab NCOMMS 2019), mTOR activation and metabolism (Walzer Lab, ELIFE 2017 and Lee Lab, JACIE 2018), with bearing on NK cell effector function. I realize that it is not realistic to redo all the experiments using shorter training periods (for example 48 hours) but it would be very exciting to see the outcome of experiments performed before major effects of metabolic and transcriptional reprogramming have taken place and before onset of cell proliferation. The authors may want to comment on whether it may be feasible to study the impact of training with self ligands under more homeostatic conditions and possibly discuss this weakness.

These are excellent points. We now address the potential weakness of our experimental design in the discussion section as follows:

“A potential weakness of our experimental design is that culturing NK cells for several days with cytokines induces a variety of transcriptional and metabolic changes that could confound the detection of NK cell education. Such changes include lysosomal biogenesis (Goodridge et al., 2019) and alterations in metabolism via the mTOR pathway (Almutairi et al., 2019; Marcais et al., 2017). Induction of these cellular programs also likely contributes to donor-specific variation of cell cultures.”

(lines 547-552)

4) In the results section, the authors provide evidence demonstrating that the observed effects are not "simply" based on more activation of NK cells exposed to targets expressing self ligands during the training phase (Fig 2F). While I see the importance of excluding effects on cell survival and proliferation, the point dealing with activation is more complex, since it may in fact be the opposite effect (disarming) we are seeing during the training phase? Indeed, somewhat confusingly, in the discussion the authors focus on this latter effect and refute disarming as an explanation for the observed in vitro education.

One would expect that cells exposed to self ligands should be less activated during the training phase. In fact, this is a scenario we anticipate during in vivo education (although at much lower cytokine concentrations). I believe the data presented are compatible with the disarming model of NK cell education. Priming under self inhibition would protect from disarming. Such potential protection from disarming would have to be monitored during the training phase, corresponding to the no test (no target) conditions at various time-points after training. Although I agree with the authors that one would possibly expect the training effect to be lost after exposure to K562 in the test phase, I do not think there is sufficient evidence to totally refute disarming as an underlying principle behind the relative increase in function in cells that are inhibited during the priming phase. One possible reason for such a preserved effect could be more long-term changes in metabolism in cells that are primed in IL-2 under inhibition versus those are not.

The argument made about the superior stimulatory effect of any feeder (carrying self and non-self ligands) versus cytokine alone is not an argument against disarming since feeders induce a much stronger proliferative response and the two conditions are not possible to compare. Cytokine stimulation alone has been shown to dampen NK cell functionality through effects on metabolism.

I guess what I am trying to say that manipulating/controlling such disarming events in vitro would be of no less importance (and would still be referred to as education) than any other unknown mechanism, operating during the in vitro training. Please consider softening the strong statements against this alternative (eg disarming) explanation.

We see the reviewer's point regarding the disarming model of NK cell education. As a result, we have addressed the disarming model directly in our discussion:

“One mechanistic interpretation of our data is that NK cells receiving inhibitory signaling during the training phase were able to conserve functionality for the testing phase. In this scenario, KIR3DL1⁺ NK cells in the Bw4⁻ training condition became more activated during training than those in the Bw4⁺ condition, thus depleting cellular resources. Then, in the testing phase, NK cells trained with Bw4⁺ had more resources available to respond to missing-self. Such an interpretation fits with the disarming model of NK cell education, in which a lack of inhibitory signaling erodes NK cell functionality over time (Raulet and Vance, 2006).”

(lines 582-589)

Moreover, our discussion now embraces the reviewer's excellent point regarding the unfairness of comparisons between mono-cultured and co-cultured samples, concluding that the disarming model remains one reasonable interpretation of our data:

“A possible contradiction of this model is that NK cells cultured alone should conserve the most resources. Yet, untrained NK cells in our experiments did not outperform NK cells trained by Bw4. However, because of the extensive

reprogramming induced by target interaction, it is unrealistic to make direct comparisons between trained and untrained NK cells. Therefore, the disarming model remains one reasonable interpretation of our data.”

(lines 591-595)

5) It is interesting to see that the authors can detect differences in granzyme B levels, in keeping with the study of Goodridge et al. However, effects of priming are important to consider in this context since IL-2 leads to a 100-1000 fold increase in granzyme B levels compared to baseline. Thus, granzyme B is likely a poor proxy of education under conditions of cytokine priming, in particular in the absence of a rest period. Would it be helpful to calculate the ratio of granzyme B expression between self versus non-self KIR+ NK cells to monitor how this change after 5 days in IL-2 and how is this changed by the training? Should be possible to extract from the present data and compared to baseline levels without priming?

This was an excellent suggestion, and the requested data is shown in Figure 6D. This comparison of the KIR3DL1⁺ / KIR2D⁺KIR3DL1⁻ ratio of granzyme B ultimately became the most robust evidence of increased granzyme B due to in vitro education in our manuscript, resulting in a significant statistical effect with only four randomly selected Bw4- donors.

6) Typo in figure legends Figure 2. Something is missing. "donors"?

The figure referenced legend has been updated and corrected.

7) Figure 3E/4B: Replace no testing with no target?

We have altered the figures and terminology as suggested throughout the manuscript.

Reviewer #3:
Major Points:

1. The results section is tedious and should be shortened considerably. One suggestion is to remove text from the results section which duplicates what is in the methods section, or figure legends.

We have taken the reviewer's suggestion to remove duplicate information. As a result, despite now including additional data comprising 2 new figures, the text of the results section has been reduced from ~28,000 characters to ~21,000 characters.

2. Stats for Bw4+ donors should be included in results shown since differences are discussed. In several cases, data obtained for Bw4+ donors is not even presented.

We have inserted statistics for Bw4⁺ donors to match all shown statistics of Bw4⁻ donors, and include data for Bw4⁺ donors in all cases in which they were part of the experimental design. These include the experiments shown in Figures 3, 4, 5, 6, and 7.

3. Since it's challenging to compare MFI values in different experiments due to day to day variations in sample processing and cytometer performance, especially for cytometers using PMTs, it would help to clarify (Methods section) that the same Fortessa cytometer was used in both experiments with similar cytometer performance.

In all cases, we collected each of the data points that appear in paired comparisons on the same day using the same cytometer. We have confirmed that this was done in the methods section as follows:

“Cells were analyzed using a LSR II flow cytometer (BD) at the Stanford Shared FACS Facility (Stanford, CA). Sphero Mid-range Rainbow Fluorescent beads (Spherotech, Lake Forest, IL) were used to calibrate and verify the performance of all channels prior to each data collection session. The same cytometer was used when comparing samples from the same experiment collected on different days. Data points appearing in paired comparisons were collected on the same cytometer in the same day without interruption.”

(lines 780-785)

4. Histograms are useful for presenting and comparing MFI values for two populations and should be included along with representative gates for all positive fractions. As these data naturally have a large amount of variation, it may help to concatenate whatever number of down sampled events from the positive GZMB fraction for each individual sample within an experiment to generate a representative graph.

This is an excellent suggestion. We agree that histograms can provide a more intuitive view of the MFI data. We concatenated our .fcs files as suggested. These histograms are now shown in Figures 6C and 6E, supporting our original findings.

5. Presentation of the details of an approach or experimental setup in some cases is unclear. As one example, development of a synthetic product to educate NK cells is first described using anti-CD2-DNAM1 beads. Quickly this changed to use of anti-CD2-NKp46 beads for most of the experiment for some unknown reason. Why? And then data in Fig 5 indicate anti-CD2 is sufficient to trap HLA molecules. So, why were further studies then performed using anti-CD2-DNAM1? This is confusing, and not explained. Related to this, Supp Fig 3c and its legend are in disagreement about what type of beads were actually used in the experiment.

We agree with the reviewer's point regarding the potential confusion of using different types of beads in the experiments. We have replaced the data in Figure 7, which now consistently uses anti-CD2-DNAM1 beads throughout figures 7A and 7C. Beads coated with a single antibody clone are now only used in Fig. 7B, which is required to experimentally determine whether anti-CD2 or anti-DNAM1 mediates reverse trogocytosis.

Minor Points:

1. The anti-KIR Ab therapy discussed on p5 has failed in clinical trials.

We now allude to this result in mentioning this type of intervention:

“Strategies for boosting the response of NK cells to cancer include blocking KIR-HLA binding in vivo, which theoretically promotes NK cell activation (Kim and Kim, 2018). However, KIR blockade interventions have thus far not been successful in clinical trials. This may be partly because they are only applicable to the degree that the patient's NK cells have been educated in vivo through the targeted KIR.”

(lines 98-103)

2. It may help to include discussion of the reasoning for chosen time points, and whether other training time intervals were considered or tested in developing this NK cell education assay.

We now discuss this parameter as suggested:

“In developing the training protocol, we found that the timing of training was dependent on the training cell line used, and that inconsistent results were obtained if NK cells were trained for periods of only 24 or 48 hours. These observations therefore imply that the signaling necessary for in vitro education occurs at the outset of training, and is followed by several days of maturation.”

(lines 509-513)

3. The B&EF equation should be included in the methods section and then briefly mentioned in the results.

In the results section we now refer the reader to methods section for a description of the B&EF equation, as follows.

“To examine this possibility, we calculated a relative binding and expression factor (B&EF) using the KIR3DL1 and HLA-B types of the donors as well as the HLA-B types of the Bw4⁺ BCL (see Methods). The B&EF value reflects the strength of the KIR3DL1-Bw4 interaction during training, as well as the average surface expression of KIR3DL1 for each donor.”

(lines 291-295)

In the Materials and Methods section we now describe the B&EF equation under its own heading:

“Binding and Expression Factor (B&EF) Calculation. Binding scores were generated by color quantifying the heatmap dataset of Boudreau et al using Adobe Photoshop CS6 Ver. 16 (Adobe Systems, Inc.). Binding quantities were normalized on a continuous scale from 1-10. Donors for whom all KIR-HLA-B combinations were represented in the binding data were selected. Each of the four possible HLA-B-KIR bonds when training with the Bw4⁺ cell line was calculated for each donor. Scores for the four bonds were then averaged, resulting in a single overall binding score for each donor. The overall binding score for each donor was then multiplied by 1000 and multiplied by the MFI of KIR3DL1 for that donor. The KIR3DL1 MFI was recorded after NK cell isolation but prior to training or any co-culture.”

(lines 789-798)

4. Box and whisker plots with SD or SEM may aid interpretation of the data. Additionally, the connector lines for training data in 2 different conditions shown in multiple figures are barely visible.

We have changed the data plots throughout the manuscript to feature mean with standard deviation, overlaid on individual values and connector lines where applicable. We agree that this provides the reader more information, and we also hope this new aesthetic is more visually appealing. Additionally, connector lines have been made a darker color throughout the manuscript.

5. Why show Fig 3E stats comparing what happened for Bw4- donor NK cell training to what happened for Bw4+ donor NK cell training?

The indication of statistics in this plot, now Fig. 4E, have been changed. This plot now features data from combined experiments, which has altered the statistical results. The statistics shown are now consistent with the other plots in the manuscript.

6. The use of a table to present data (especially for ratios) described in the results section may help to efficiently present and describe the main findings.

We thank the reviewer for this excellent suggestion. The manuscript now includes a Table as suggested, which we believe has improved reporting clarity considerably. This table is referenced throughout the results section that describes the data in Fig. 3.

7. No CD107a expression data is shown in supp Fig 1 so this mention should be removed from the legend.

This data now appears in Fig. 2B, and with appropriate references in the figure legend.

8. Discussion text on P27 describing the conservation of potency and a theoretical activation disadvantage is almost impenetrable.

We have revised this section of the discussion to be more clear. In line with this request and that of another reviewer, we now use the disarming model of NK cell education as the basis for our discussion:

“One mechanistic interpretation of our data is that NK cells receiving inhibitory signaling during the training phase were able to conserve functionality for the testing phase. In this scenario, KIR3DL1⁺ NK cells in the Bw4⁻ training condition became more activated during training than those in the Bw4⁺ condition, thus depleting cellular resources. Then, in the testing phase, NK cells trained with Bw4⁺ had more resources available to respond to missing-self. Such an interpretation fits with the disarming model of NK cell education, in which a lack of inhibitory signaling erodes NK cell functionality over time (Raulet and Vance, 2006).”

(lines 582-589)

9. Manuscript should be carefully proofread for typos. Some are listed here, but there are others.

- P7 - rhIL2

- *The use of 500IL2+IL12+Bw4+BCL is odd since the cells are not IL2+ or IL12+. It would seem simpler to refer to these cells as cytokine-induced or IL2/IL12-induced.*

- *Supp Figure 1 legend (L.2) - Or[g]ange*

- *Supp Figure 2 legend (L.8) - Bromodoxyuridine*

- *Figure 2A, right panel - rhIL-2[n]*

We thank the reviewer for noting these typos. We have corrected these typos and others throughout the manuscript. Additionally, we have removed the designation “500IL2+IL12+Bw4+BCL” from any descriptions of cells. Instead, we describe the cytokine condition separately from the type of training cell in all instances. Here are some examples:

“Training with Bw4⁺BCL in high IL-2 resulted in a 33% increase in the frequency of IFN γ ⁺ cells (Table 1).”

(lines 185-186)

“Training KIR3DL1⁺ NK cells from Bw4⁺ donors with Bw4⁺BCL and IL-12 did not improve their degranulation (Fig. 4B, Table 1).”

(lines 213-214)

October 22, 2019

RE: Life Science Alliance Manuscript #LSA-2019-00434-TR

Prof. Peter Parham
Stanford University
Stanford University Medical Ctr.
School of Medicine
Dept. of Cell Biology
Stanford, Sherman Fairchild Bldg. D157 USA-Stanford, CA 94305

Dear Dr. Parham,

Thank you for submitting your revised manuscript entitled "In Vitro Education of Human Natural Killer Cells by KIR3DL1". As you will see, the reviewers appreciate the introduced changes and now support publication. Before sending you an official acceptance letter, please address reviewer #3's minor comments as well as the following:

- please add the supplementary figure legends in the main manuscript file
- Figure 3C is called out in the ms text, but does not exist in the figure; please fix
- please add callouts in the ms text to Fig5C and FigS1D

A. FINAL FILES:

-- Summary blurb (enter in submission system): A short text summarizing in a single sentence the study (max. 200 characters including spaces). This text is used in conjunction with the titles of papers, hence should be informative and complementary to the title. It should describe the context

and significance of the findings for a general readership; it should be written in the present tense and refer to the work in the third person. Author names should not be mentioned.

B. MANUSCRIPT ORGANIZATION AND FORMATTING:

Sincerely,

Andrea Leibfried, PhD
Executive Editor
Life Science Alliance
Meyerohofstr. 1
69117 Heidelberg, Germany
t +49 6221 8891 502
e a.leibfried@life-science-alliance.org
www.life-science-alliance.org

Reviewer #1 (Comments to the Authors (Required)):

I think the authors have addressed the points in a satisfactory manner, and I do not have more outstanding points.

Reviewer #2 (Comments to the Authors (Required)):

The authors have addressed all my concerns. The revised version is much improved and the reported model hold utility for investigation of the mechanism behind NK cell education.

Reviewer #3 (Comments to the Authors (Required)):

Pugh et al examined whether NK cells can be artificially educated in vitro using target cells with self-HLA class I ligands expressed, or not. The revised report provides a thorough study and compelling data for in vitro NK cell training which reliably increased the potential for NK cells to be subsequently stimulated in vitro. In vitro educated NK cells were shown to mediate missing-self killing of tumor targets and also CMV-infected target cells which poorly express cell surface class I molecules due to CMV downregulation. In addition, the report shows that beads conjugated to anti-CD2 which have acquired HLA molecules from cell targets, can also be used to educate NK cells in vitro. The report is clearly written, concise and well organized. The main message in the revised manuscript is effectively and fairly presented with adequate and appropriate referencing of published data. The observations and conclusions are logical and justified. The authors adequately addressed points that I raised in the previous review, and they seemingly have addressed the comments raised by the other referees. The report has been strengthened significantly. Overall this is an intriguing study as it provides a cornerstone to begin dissecting the molecular basis of NK cell education. Several lingering very minor issues are below.

1. The W6/32 Ab does not appear to be included in the materials and methods section, nor the purchase / production of F(ab')₂ molecules.
2. L. 242 - Data is shown in Supp Fig 1C, I think. L. 244 - Data shown in Supp Fig 1D.
3. L. 268 -271 - The authors should consider citing Babic et al (2010) JEM 207:2663 which showed NK "missing-self" recognition of (delta)m04 MCMV infected targets.
4. L. 561 - typo

November 4, 2019

RE: Life Science Alliance Manuscript #LSA-2019-00434-TRR

Prof. Peter Parham
Stanford University
Stanford University Medical Ctr.
School of Medicine
Dept. of Cell Biology
Stanford, Sherman Fairchild Bldg. D157 USA-Stanford, CA 94305

Dear Dr. Parham,

Thank you for submitting your Research Article entitled "In Vitro Education of Human Natural Killer Cells by KIR3DL1.". It is a pleasure to let you know that your manuscript is now accepted for publication in Life Science Alliance. Congratulations on this interesting work.

DISTRIBUTION OF MATERIALS:

Again, congratulations on a very nice paper. I hope you found the review process to be constructive and are pleased with how the manuscript was handled editorially. We look forward to future exciting submissions from your lab.

Sincerely,
